# Neutrophil-induced ferroptosis promotes tumor necrosis in glioblastoma progression

Patricia P. Yee [1,2], Yiju Wei[1], Soo-Yeon Kim [1], Tong Lu[1], Stephen Y. Chih [2], Cynthia Lawson [3], Miaolu Tang[1], Zhijun Liu[1], Benjamin Anderson [1], Krishnamoorthy Thamburaj[4], Megan M. Young[1,5], Dawit G. Aregawi[6,7,8], Michael J. Glantz[6,7,8], Brad E. Zacharia[6,7,9], Charles S. Specht[3,6,10], Hong-Gang Wang[1,5] & Wei Li [1,11✉]

Tumor necrosis commonly exists and predicts poor prognoses in many cancers. Although it is thought to result from chronic ischemia, the underlying nature and mechanisms driving the involved cell death remain obscure. Here, we show that necrosis in glioblastoma (GBM) involves neutrophil-triggered ferroptosis. In a hyperactivated transcriptional coactivator with PDZ-binding motif-driven GBM mouse model, neutrophils coincide with necrosis temporally and spatially. Neutrophil depletion dampens necrosis. Neutrophils isolated from mouse brain tumors kill cocultured tumor cells. Mechanistically, neutrophils induce iron-dependent accumulation of lipid peroxides within tumor cells by transferring myeloperoxidase-containing granules into tumor cells. Inhibition or depletion of myeloperoxidase suppresses neutrophil-induced tumor cell cytotoxicity. Intratumoral glutathione peroxidase 4 over-expression or acyl-CoA synthetase long chain family member 4 depletion diminishes necrosis and aggressiveness of tumors. Furthermore, analyses of human GBMs support that neutrophils and ferroptosis are associated with necrosis and predict poor survival. Thus, our study identifies ferroptosis as the underlying nature of necrosis in GBMs and reveals a pro-tumorigenic role of ferroptosis. Together, we propose that certain tumor damage(s) occurring during early tumor progression (i.e. ischemia) recruits neutrophils to the site of tissue damage and thereby results in a positive feedback loop, amplifying GBM necrosis development to its fullest extent.

[1] Division of Hematology and Oncology, Department of Pediatrics, Penn State College of Medicine, Hershey, PA, USA. [2] Medical Scientist Training Program, Penn State College of Medicine, Hershey, PA, USA. [3] Division of Anatomic Pathology, Department of Pathology and Laboratory Medicine, Penn State College of Medicine, Hershey, PA, USA. [4] Department of Radiology, Penn State College of Medicine, Hershey, PA, USA. [5] Department of Pharmacology, Penn State College of Medicine, Hershey, PA, USA. [6] Division of Neuro-Oncology and Skull Base Surgery, Department of Neurosurgery, Penn State College of Medicine, Hershey, PA, USA. [7] Penn State Cancer Institute, Penn State College of Medicine, Hershey, PA, USA. [8] Department of Medicine, Penn State College of Medicine, Hershey, PA, USA. [9] Department of Otolaryngology-Head and Neck Surgery, Penn State College of Medicine, Hershey, PA, USA. [10] Department of Neurology, Penn State College of Medicine, Hershey, PA, USA. [11] Department of Biochemistry and Molecular Biology, Penn State College of Medicine, Hershey, PA, USA. ✉email: weili@pennstatehealth.psu.edu

Tumor necrosis is a histopathological term used to generally describe tissue death in cancers. It is a common feature and poor prognostic predictor in a variety of advanced cancers, such as invasive breast cancer[1], non-small cell lung cancer[2], malignant mesothelioma[3], clear cell renal cell carcinoma[4], malignant gastrointestinal stromal tumors[5], Ewing's sarcoma of the bone[6], endometrial cancer[7], and glioblastomas (GBMs)[8]. Necrosis in GBM, the deadliest and most common primary adult brain malignancy, is of particular interest since it is a diagnostic hallmark and positively correlates with tumor aggressiveness and poor outcomes[8–10]. Like other solid tumors, GBMs are characterized by extensive tissue hypoxia, likely due to rapid tumor expansion outstripping vascular supply[11]. It is thought that hypoxia serves as the initial trigger of necrosis, instigating cell death by precluding oxidative phosphorylation and abruptly depleting intracellular ATP. Although ATP depletion may disrupt a series of intracellular homeostatic processes and lead to uncontrolled cell death, the nature of tumor necrosis remains ill-defined, and the molecular mechanisms underlying its development remain unclear.

Damaged tissues, described histologically as necrosis, are immunogenic. They release damage-associated molecular patterns (DAMPs), such as high-mobility group protein B1 (HMGB1). DAMPs often act as chemoattractants to recruit neutrophils, the most abundant immune cell in human blood, to the site of necrosis[12]. Although traditionally considered a homogeneous and short-lived population, it is now appreciated that neutrophils exhibit heterogeneous immunophenotypes with dynamic functional plasticity. Beyond their ties to tissue damage, neutrophils are well-known to be present in various types of cancers as so-called tumor-associated neutrophils (TANs). Their strong correlation with a variety of advanced cancers has attracted a long-standing interest in exploring neutrophils as clinical biomarkers and therapeutic targets. However, the roles of neutrophils in cancer remain controversial[13–15]. On one hand, recent studies suggested that neutrophils can promote tumor progression by releasing factors to promote remodeling of the extracellular matrix, stimulate angiogenesis, and modulate other inflammatory cells. Factors released by neutrophils can also directly promote tumor cell proliferation and invasion. On the other hand, neutrophils can also play anti-tumor roles by virtue of their cytotoxicity. Studies of neutrophil-triggered cytotoxicity under different contexts suggested that neutrophil-derived reactive oxygen species (ROS), tumor necrosis factor-related apoptosis-inducing ligand (TRAIL), or hydrogen peroxide is responsible for neutrophil-mediated tumor cell-killing[16,17]. Nevertheless, the mechanisms by which the killing is accomplished, the type of cell death, and the implications for tumor progression remain elusive.

Ferroptosis is a type of iron-dependent regulated cell death mediated by lethal accumulation of lipid peroxides[18]. Disruption of glutathione-dependent lipid peroxide reduction systems leads to the accumulation of lipid-based ROS. This can be triggered by disrupting the function of the system Xc⁻ cystine/glutamate antiporter or glutathione peroxidase 4 (GPX4)[19]. Because of cancer cells' high iron levels and their increased sensitivity to ferroptosis induction, ferroptosis has been proposed to be promising for cancer therapeutics[20]. Consistent with this notion, expression of tumor suppressors, such as p53 or BAP1, in cancer cells can inhibit tumorigenesis by inducing ferroptosis[21,22]. However, p53 expression can also inhibit ferroptosis in some cancer cells[23,24], suggesting the role of ferroptosis in cancer could be complex and deserves further investigation.

In this study, we investigate the roles of neutrophils in GBM tumor necrosis using an orthotopic xenograft GBM mouse model faithfully recapitulating the extent of necrosis observed in GBM patients. This model exhibits the temporal and spatial correlation between necrosis and neutrophils. We demonstrate that neutrophil depletion reduces GBM necrosis and that TANs isolated from GBM tumor-bearing mice, as well as granulocyte colony-stimulating factor (G-CSF) activated mature neutrophils induce tumor cell death when cocultured in vitro. Mechanistically, such neutrophil-mediated tumor cell killing is achieved via ferroptosis and relies on intercellular transfer of neutrophil-specific granules into tumor cells. The granules contain myeloperoxidase (MPO), whose activity is required for the tumor cell killing. Suppressing ferroptosis by manipulating key ferroptosis regulators in tumor cells, such as GPX4 and acyl-CoA synthetase long chain family member 4 (ACSL4), reduces necrosis and dampens tumor aggressiveness. Finally, we provide evidence that neutrophils and ferroptosis are also associated with tumor necrosis and poor survival in GBM patients.

## Results

**Hyperactivating TAZ promotes GBM mesenchymal (MES) transition and tumor necrosis.** Transcriptional coactivator with PDZ-binding motif (TAZ) is an oncogenic transcriptional regulator in the Hippo pathway. It was shown that the TAZ-driven transcriptional program is associated with GBM MES differentiation and aggressive progression[25]. To examine if *TAZ* expression is increased in the MES subtype of GBM, we analyzed the TCGA GBM dataset through cBioPortal (www.cbioportal.org). More tumors of MES subtype show higher *TAZ* expression than those of proneural (PN) or classical (CL) subtypes (Fig. 1a). To study how TAZ activation drives aggressive GBM progression, we devised a TAZ-driven xenograft GBM mouse model by stably expressing a constitutively active TAZ mutant (TAZ$^{4SA}$)[26] in a commonly used LN229 human GBM cell line (Supplementary Fig. 1a), which contains a P98L missense mutation in p53 (Cancer Cell Line Encyclopedia). Mice intracranially implanted with TAZ$^{4SA}$-expressing tumor cells (hereafter denoted LN229$^{TAZ(4SA)}$) showed significantly shorter survival than those implanted with vector-transduced tumor cells (hereafter denoted LN229$^{vector}$) (Fig. 1b). LN229$^{TAZ(4SA)}$ tumors grow much faster than LN229$^{vector}$ tumors (Supplementary Fig. 1b). These results were consistent with previous observations[27] and suggested that the former tumors are more aggressive than the latter ones. Blotting the tumor lysates for MES markers (fibronectin, CD44, and CTGF) revealed that LN229$^{TAZ(4SA)}$ tumors express these proteins at higher levels, suggesting a MES transformation in vivo (Fig. 1c). Histological studies found that LN229$^{TAZ(4SA)}$ tumors are much more heterogeneous than LN229$^{vector}$ tumors and contain large areas of necrosis, whereas LN229$^{vector}$ tumors do not develop detectable necrosis (Fig. 1d–f). Notably, such a difference existed even when LN229$^{TAZ(4SA)}$ and LN229$^{vector}$ tumors were examined at the same size (Supplementary Fig. 1c), suggesting that tumor size does not determine the presence or absence of tumor necrosis. Since heterogeneity and extensive necrosis are common features of GBMs, this histological appearance suggested that TAZ hyperactivation drives tumor progression.

As a transcriptional coactivator, TAZ regulates gene transcription by interacting with the transcription factor TEAD[28]. To test whether the aggressiveness of LN229$^{TAZ(4SA)}$ tumors depends on this transcriptional activity, we used another TAZ mutant, TAZ$^{(4SA-S51A)}$, which is unable to bind to TEAD[28]. Tumors of LN229 cells expressing TAZ$^{(4SA-S51A)}$ (hereafter denoted LN229$^{TAZ(4SA-S51A)}$) are more similar to LN229$^{vector}$ than LN229$^{TAZ(4SA)}$ tumors, although some LN229$^{TAZ(4SA-S51A)}$ tumors can develop small necrosis (Fig. 1d, f). Notably, TAZ$^{(4SA-S51A)}$ and TAZ$^{(4SA)}$ were expressed at

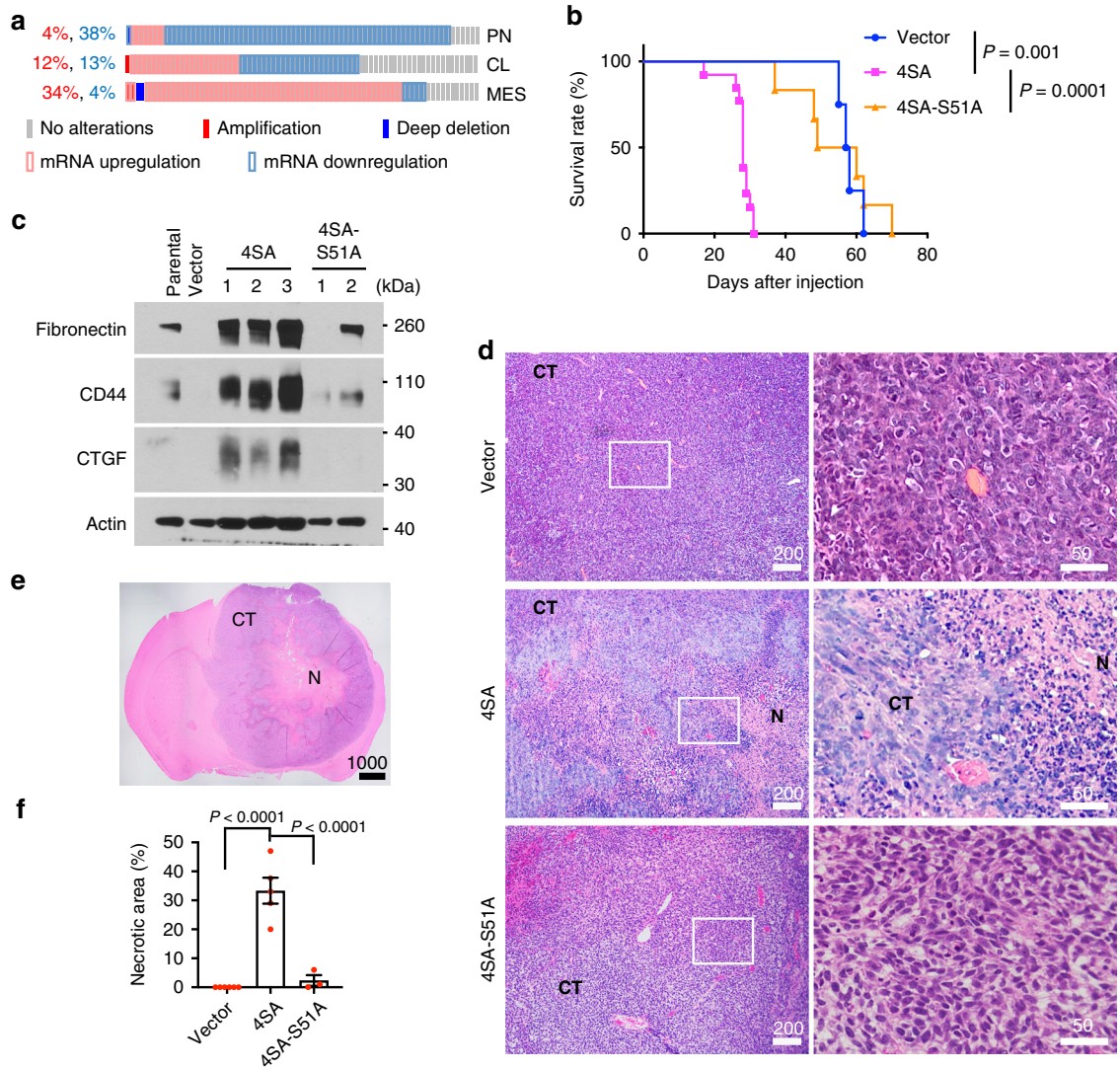

**Fig. 1 Hyperactivating TAZ promotes GBM MES transition and tumor necrosis. a** The TCGA GBM dataset (Provisional, $n = 540$ tumors) was grouped into proneural (PN; $n = 166$ tumors), classical (CL; $n = 204$ tumors) and mesenchymal (MES; $n = 170$ tumors) subtypes as defined previously[79]. *TAZ* expression in each subtype was examined through cBioPortal using U133 microarray only. The *z*-score threshold was ± 1. **b** Kaplan–Meier survival curves of mice implanted with LN229 cells stably transduced with empty vector control ($n = 4$ mice), TAZ[4SA] ($n = 13$ mice), or TAZ[4SA-S51A] ($n = 6$ mice). Two-sided log-rank test. **c** Immunoblot of MES markers, including Fibronectin 1, CD44, and connective tissue growth factor (CTGF) in whole-tumor lysates obtained from mice implanted with LN229[parental] ($n = 1$ tumor), LN229[vector] ($n = 1$ tumor), LN229[TAZ(4SA)] ($n = 3$ tumors), or LN229[TAZ(4SA-S51A)] ($n = 2$ tumors). **d** Representative H&E-stained formaldehyde-fixed paraffin-embedded sections obtained from mice implanted with LN229[vector] ($n = 6$ tumors), LN229[TAZ(4SA)] ($n = 5$ tumors), or LN229[TAZ(4SA-S51A)] ($n = 3$ tumors) reaching endpoints at low (left) and high (right) magnifications. Indicated numbers of animals ($n$) from each condition were examined independently with consistent observations. **e** Representative H&E-stained formaldehyde-fixed paraffin-embedded section of a typical LN229[TAZ(4SA)] tumor-bearing brain with clear central tumor necrosis (denoted by N) and cellular tumor (denoted by CT). $n = 5$ tumors. The number of animals examined independently in each condition is listed above; observations were consistent within each group. **f** Necrosis-to-tumor area ratios comparison among mice implanted with LN229[vector] ($n = 6$ tumors), LN229[TAZ(4SA)] ($n = 5$ tumors), or LN229[TAZ(4SA-S51A)] ($n = 3$ tumors) using 3–4 step H&E-stained sections (each 50 μm apart) obtained from the same tumor; the average of these ratios was used to represent each tumor. Numerical data are presented as mean ± s.e.m. Each data point represents an animal. Ordinary one-way ANOVA. All scale bars are in μm. Source data are provided as a Source Data file.

similar levels (Supplementary Fig. 1a). This result indicated that interaction with TEAD is necessary for TAZ to drive such malignant transformation. To examine if the effects of TAZ hyperactivation are limited to LN229 cells, we used the same strategies with two additional commonly used human GBM cell lines, U87MG and LN18. While U87MG does not contain a p53 mutation, LN18 contains a C238S missense mutation in p53 (Cancer Cell Line Encyclopedia). Similar to LN229 cells, TAZ hyperactivation in these cells also resulted in aggressive GBMs containing extensive necrosis (Fig. 1d and Supplementary Fig. 1d–g). These results suggested that

the TAZ-TEAD-mediated transcriptional program drives GBM MES differentiation and progression, which are accompanied by extensive tumor necrosis.

**GBM necrosis is extensively infiltrated with tumor-associated neutrophils (TANs).** Although the necrotic areas in LN229[TAZ(4SA)] tumors generally have low cellularity, a close examination revealed that these areas contained certain cells with smaller and denser nuclei than typical tumor cells (Fig. 2a, arrowheads vs.

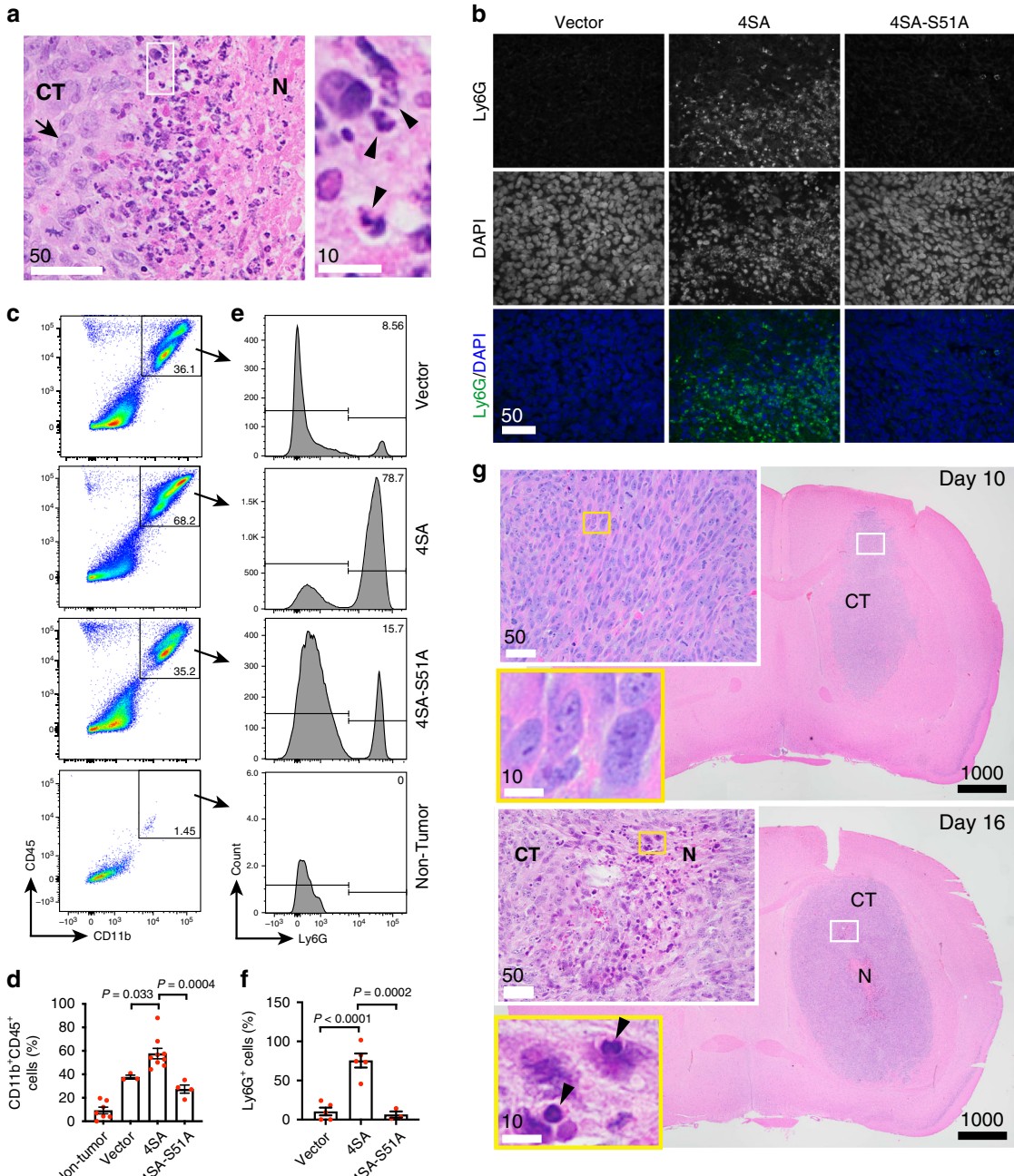

**Fig. 2 GBM necrosis is extensively infiltrated with neutrophils. a** Representative high-magnification H&E-stained section obtained from a LN229[TAZ(4SA)] tumor-bearing mouse ($n = 3$ tumors) showing areas bordering cellular tumor (CT) and necrosis (N). The outlined area is enlarged and shown on the right. Three animals from each condition were examined independently with consistent observations. **b** Representative immunofluorescent staining of mouse-specific neutrophil marker, Ly6G, and DAPI staining on paraffin-embedded sections of mice implanted with LN229[vector], LN229[TAZ(4SA)], or LN229[TAZ(4SA-S51A)] ($n = 3$ tumors for each) upon reaching endpoints. Three animals from each condition were examined independently with consistent observations. **c** Representative flow cytometry analysis of murine myeloid cell markers CD11b and CD45 on cells isolated from tumor tissues of mice implanted with LN229[vector], LN229[TAZ(4SA)], or LN229[TAZ(4SA-S51A)] upon reaching endpoints. Cells isolated from a non-tumor-bearing mouse served as a baseline. **d** Percentage of CD11b+CD45+ cells from flow cytometry analyses shown in **c**; LN229[vector] ($n = 3$ tumors), LN229[TAZ(4SA)] ($n = 9$ tumors), LN229[TAZ(4SA-S51A)] ($n = 4$ tumors), and non-tumor-bearing mice ($n = 7$ mice). Ordinary one-way ANOVA. **e** Further immunophenotyping of CD11b+CD45+ cells shown in **c** via Ly6G flow cytometry, with representative results shown here. **f** Percentage of Ly6G+ cells (a.k.a. neutrophils) shown in **e**; LN229[vector] ($n = 5$ tumors), LN229[TAZ(4SA)] ($n = 5$ tumors), and LN229[TAZ(4SA-S51A)] ($n = 3$ tumors). Ordinary one-way ANOVA. **g** Representative low- and high-magnification H&E-stained formaldehyde-fixed paraffin-embedded brain sections of mice implanted with LN229[TAZ(4SA)] collected 10 days after tumor implantation (top; $n = 3$ tumors) and 16 days after tumor implantation (bottom; $n = 3$ tumors). N denotes central tumor necrosis and CT denotes cellular tumor. Three animals from each condition were examined independently with consistent observations. Outlined areas are enlarged and shown in the color-coded panels. **a–g** $n$ indicates total number of animals. Numerical data are presented as mean ± s.e.m. Each data point represents an animal. All scale bars are in μm. Source data are provided as a Source Data file.

arrows). These cells are likely dead tumor cells or infiltrated immune cells. Most of these cells contain multi-lobed nuclei, a characteristic of neutrophils (Fig. 2a, arrowheads). To confirm their identity, we performed immunohistochemistry on tumor sections using a murine neutrophil marker, Ly6G[29]. Ly6G-positive cells are readily observed in the necrotic areas of LN229[TAZ(4SA)] tumors but rarely found in LN229[vector] or LN229[TAZ(4SA-S51A)] tumors (Fig. 2b). We then used flow cytometry to verify this observation quantitatively. Neutrophils are terminally differentiated leukocytes derived from the myeloid lineage. Therefore, we first used myeloid cell markers, CD11b and CD45, to gate out CD11b$^+$CD45$^+$ myeloid cells. Compared to non-tumor-bearing mouse brains, all tumors were infiltrated with significantly more myeloid cells (Fig. 2c, d). Such infiltration of myeloid cells was more prominent in LN229[TAZ(4SA)] tumors than LN229[vector] or LN229[TAZ(4SA-S51A)]. Next, we used a murine neutrophil marker, Ly6G, to further verify whether these tumor-infiltrating CD11b$^+$CD45$^+$ myeloid cells were indeed neutrophils. Unlike the other two types of tumors, Ly6G$^{high}$ cells appear to be the major population of infiltrating myeloid cells in LN229[TAZ(4SA)] tumors (Fig. 2e, f). Collectively, these results demonstrated a marked increase of neutrophil infiltration in LN229[TAZ(4SA)] tumors.

As neutrophils were spatially correlated with the necrosis, especially at the interfaces of cellular tumor and necrotic areas (Fig. 2a, b), we sought to examine if a temporal correlation between neutrophils and necrosis also exists. First, we used CD11b and CD45 to examine myeloid cells in LN229[TAZ(4SA)] tumors at different stages of tumor progression. Flow cytometry indicated that CD45$^+$ cells (i.e., infiltrating mouse immune cells) in tumors at day 20 after tumor implantation can be separated into three major populations based on CD11b and CD45 signal intensities, which we named CD11b$^{high}$CD45$^{high}$, CD11b$^{med}$CD45$^{med}$, and CD11b$^{low}$CD45$^{low}$ cells (Supplementary Fig. 2a). At this stage, the tumor-infiltrating immune cells consist of nearly equal proportions of the three cell populations. As tumors grow, the CD11b$^{high}$CD45$^{high}$ cells gradually become the dominant population (Supplementary Fig. 2a, b). Previous studies reported that microglia in inflamed brains can be distinguished from peripherally-infiltrating macrophages based on lower microglial CD45 expression[30]. However, CD45 expression in neutrophils relative to microglia and macrophages in the brain was unclear. To examine which cell population contains neutrophils, we used the murine neutrophil marker Ly6G. The CD11b$^{high}$CD45$^{high}$ population largely consisted of Ly6G$^+$ cells, whereas the other two populations essentially lack Ly6G$^+$ cells (Supplementary Fig. 2a). Such specific enrichment of Ly6G$^+$ cells does not change during tumor development (Supplementary Fig. 2c). Since both Ly6G$^+$-enriched CD11b$^{high}$CD45$^{high}$ cell population and necrosis together become more prominent during tumor development, the results supported that neutrophils and necrosis are temporally correlated.

To further examine the correlation between neutrophils and necrosis at an earlier stage, we used immunohistochemistry. A brain tumor was visible on hematoxylin and eosin (H&E) staining at day 10 after tumor cell implantation. However, there was no detectable tumor necrosis (Fig. 2g). These tumors were as homogeneous as LN229[vector] tumors (Figs. 2g vs. 1d). Very few neutrophils were seen in these tumors. At day 16 after tumor cell implantation, the derived tumors contained a few necrotic foci that were infiltrated with neutrophils (Fig. 2g). Notably, H&E staining (Fig. 2g, insets) and Ly6G immunohistochemistry (Supplementary Fig. 2d) on these day-16 tumors revealed neutrophil accumulation in certain areas where tissues appeared slightly damaged. These results suggested that neutrophil infiltration may occur in areas of developing neo-necrosis.

In summary, these results showed that GBM necrosis is closely associated, both spatially and temporally, with neutrophil infiltration.

**Neutrophils facilitate GBM necrosis in vivo and induce tumor cell death ex vivo.** It is known that damaged tissues can release DAMPs, which are able to recruit neutrophils. It is thus unsurprising to see spatial and temporal correlation between GBM necrosis and neutrophil infiltration in our model. However, it remains unresolved whether infiltrated neutrophils can contribute to or promote the development of tumor necrosis. To examine this, we utilized a previously established protocol to deplete neutrophils via anti-Ly6G antibody[31] and examined the impact on GBM necrosis. Of note, the anti-Ly6G antibody itself does not affect tumor cell proliferation in vitro (Supplementary Fig. 3a). We intraperitoneally administrated a monoclonal anti-Ly6G antibody or a non-specific IgG into the LN229[TAZ(4SA)] tumor-bearing mice following the indicated regimen (Supplementary Fig. 3b). Treatment courses started on day 14 after tumor cell implantation and continued until each animal reached its endpoint (~14 days), which covered the pre-necrosis stage until the terminal stage. To evaluate the effectiveness of neutrophil depletion, we first examined blood-circulating neutrophils via peripheral blood smears. Compared to the IgG-treated mice, the Ly6G-treated mice had significantly fewer circulating neutrophils throughout the whole treatment course (Supplementary Fig. 3c). The depletion appeared to be most effective after the initial treatment (e.g., days 1–6 after the first antibody administration) and then became less effective later (e.g., days 11–13 after the first antibody administration). Such depletion kinetics agree with previous studies using the same depletion strategies[32]. We then examined tumor-associated neutrophils in the brain tissues. Complete depletion was achieved at day 20 after tumor implantation (day 6 after the first antibody administration). Notably, although the effectiveness of neutrophil depletion was progressively lost, partial depletion continued (Supplementary Fig. 3d) (e.g., days 10 and 12 after the first antibody administration). At the terminal stage, tumor-associated neutrophils appeared to be largely depletion-resistant (Supplementary Fig. 3d). Since neutrophil depletion is more efficient at early stages, we examined the effect on tumor necrosis at early tumor progression (early-stage, i.e., days 20, 22, and 24 after tumor implantation) and upon reaching endpoints (terminal stage, i.e., days 26–33 after tumor implantation). Analyses of tumors from these mice showed that Ly6G-treated mice exhibited tumors containing significantly smaller necrotic cores than those from IgG-treated mice (Fig. 3a, b and Supplementary Fig. 3e). The necrosis reduction was more profound at the early-stage (62%) than terminal stage (50%). Intratumoral endothelial (CD31$^+$) cells appeared less abundant when neutrophils were depleted (Supplementary Fig. 3f). This observation is consistent with the notion that TANs promote tumor angiogenesis[33,34]. Overall, these results suggested that neutrophils are involved in promoting the development of tumor necrosis.

To directly investigate if tumor-associated neutrophils (TANs) can induce tumor cell death, we isolated TANs from the brains of LN229[TAZ(4SA)] tumor-bearing mice and cocultured them with LN229[TAZ(4SA)] cells ex vivo. In an acute (15 h) cell survival assay, cell viability was determined via luminescence. Luciferase-expressing-LN229[TAZ(4SA)] tumor cells and TANs were cocultured at ratios from 1:1 to 1:20, followed by addition of luciferin substrate and luminescence reading. As the ratio of neutrophils to tumor cells increased, fewer tumor cells survived (Supplementary Fig. 3g), suggesting that TANs are cytotoxic to tumor cells. To examine if such cell-killing ability is specific to TANs, we isolated

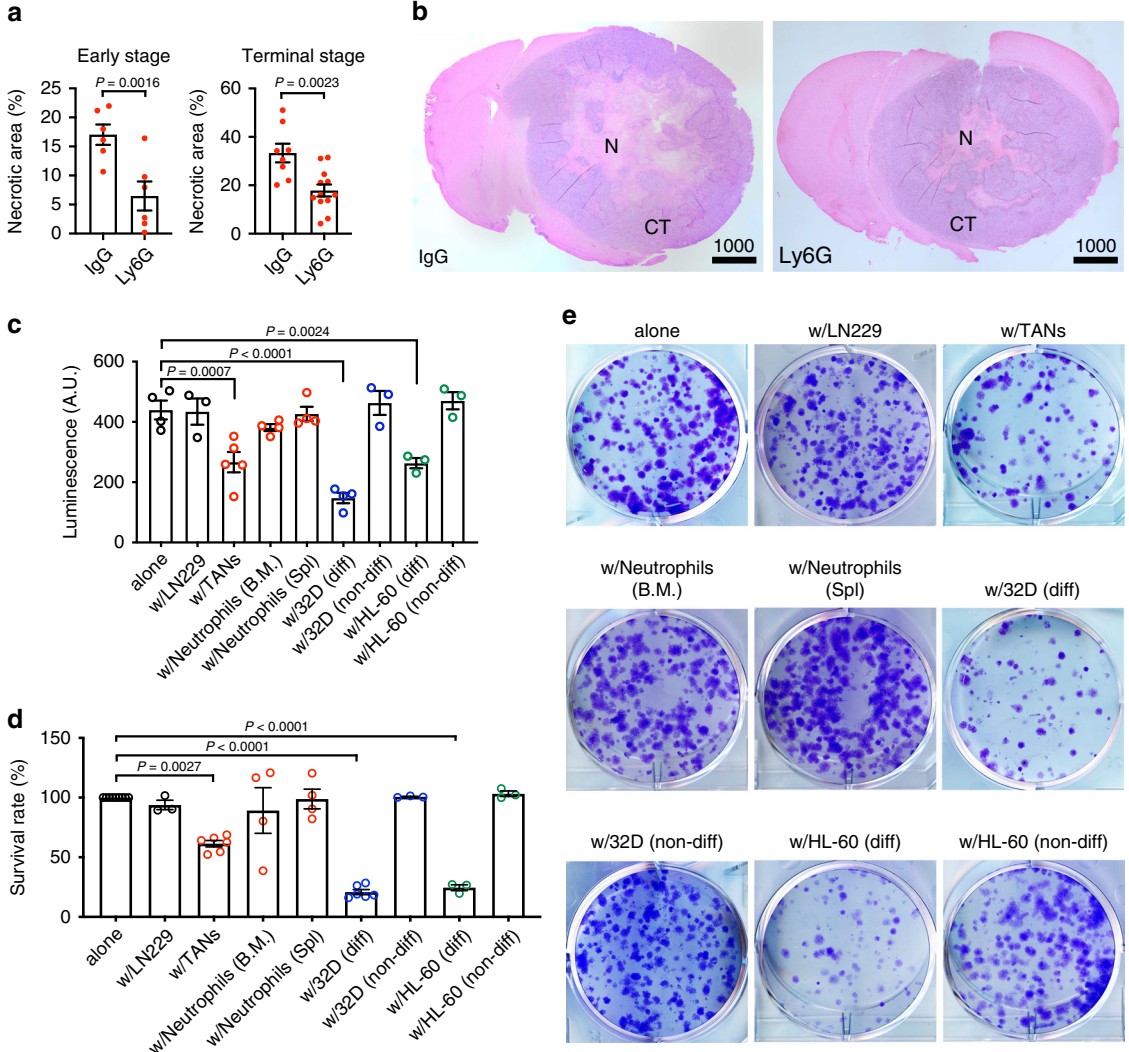

**Fig. 3 Neutrophils facilitate GBM necrosis in vivo and induce tumor cell death in vitro. a** Necrosis-to-tumor area ratios at either early tumor progression (i.e., days 20, 22, and 24 after tumor implantation) or upon reaching endpoints of LN229^TAZ(4SA) mice treated with either Ly6G or IgG isotype control ($n = 2$ for each of the three early-stage time points; $n = 12$ for terminal stage of Ly6G-treated group; $n = 8$ for terminal stage of IgG-treated group). Two-tailed paired $t$-test for early-stage and two-tailed unpaired $t$-test for terminal stage. **b** Representative H&E brain sections at endpoints as quantified in **a**; N denotes necrosis and CT denotes cellular tumor. **c** Tumor cell viability evaluated via luminescence of LN229^TAZ(4SA) cells cultured alone ($n = 4$) or with parental LN229 ($n = 3$), with TANs from LN229^TAZ(4SA) tumor-bearing mouse brains ($n = 5$), with tumor-naive neutrophils from bone marrow (B.M.) or spleen (Spl.) of non-tumor-bearing mice ($n = 4$), or with undifferentiated (non-diff) or differentiated (diff) 32Dcl3 ($n = 4$ for diff, $n = 3$ for non-diff) or HL-60 ($n = 3$ for diff, $n = 3$ for non-diff) cells. Raw luminescence levels (in arbitrary units, A.U.) are shown. Ordinary one-way ANOVA. **d** Tumor cell viability evaluated via colony-formation assays of LN229^TAZ(4SA) cells cultured alone ($n = 9$) or with parental LN229 ($n = 3$), with TANs from LN229^TAZ(4SA) tumor-bearing mouse brains ($n = 6$), with tumor-naive neutrophils from bone marrow (B.M.) or spleen (Spl.) of non-tumor-bearing mice ($n = 4$), or with undifferentiated or differentiated 32Dcl3 ($n = 6$ for diff, $n = 3$ for non-diff) or HL-60 ($n = 3$ for diff, $n = 3$ for non-diff) cells. Percent survival is normalized to LN229^TAZ(4SA) tumor cells cultured alone. Ordinary one-way ANOVA. **e** Representative whole-well images of colony-formation assay conditions as quantified in **d**. **a–d** $n$ indicates number of animals or biologically independent samples (with each sample from an independent experiment). Numerical data are presented as mean ± s.e.m. All scale bars are in μm. Source data are provided as a Source Data file.

naive neutrophils from the bone marrow and spleens of non-tumor-bearing mice. As tumor-neutrophil ratios within the LN229^TAZ(4SA) tumor appear to be much higher than 1:1, especially at the interfaces of necrotic areas and cellular tumor regions (Fig. 2a), we used the 1:20 ratio in the following in vitro assays. In the acute cell survival assay, TANs, but not tumor-naive neutrophils, induced cocultured tumor cell death (Fig. 3c). To further evaluate cell viability more directly, we used two additional approaches: flow cytometry analyses (using Sytox cell-impermeant nucleic acid fluorescent dye; Supplementary Fig. 3h) and colony-formation assays (Fig. 3d, e). These assays confirmed the specific tumor cell-killing ability of TANs. Such a

difference in killing capacity between different types of neutrophils agreed with previous reports in which tumor-associated neutrophils can exert tumor cell cytotoxicity, whereas such cytotoxicity against tumor cells is absent among neutrophils isolated from non-tumor-bearing animals[35,36]. The specific cytotoxicity of TANs, but not neutrophils from non-tumor-bearing mice, suggested that the necrosis-containing tumor microenvironment may further induce neutrophil maturation or activation to obtain such cytotoxicity.

To examine if maturation could increase neutrophil cytotoxicity, we employed established neutrophil differentiation models. It was reported that the murine myeloid cell line 32D Clone 3

(32Dcl3) can differentiate into neutrophils under induction by granulocyte colony-stimulating factor (G-CSF). The differentiated 32Dcl3 cells are functionally similar to mature neutrophils and, therefore, have been used to study neutrophil functions[37]. We examined if these in vitro differentiated neutrophils also exhibit cytotoxicity. After verifying the maturation stages of differentiated 32Dcl3 cells by flow cytometry of Ly6G[29] (Supplementary Fig. 3i, j), we tested the killing capacities of these differentiated neutrophils using the aforementioned acute viability assay. Like TANs, differentiated 32Dcl3 cells exhibit a ratio-dependent tumor cell-killing ability (Supplementary Fig. 3k). Interestingly, only the differentiated 32Dcl3 cells, but not the parental cells, induced LN229$^{TAZ(4SA)}$ tumor cell death (Fig. 3c). This was further confirmed by the Sytox dye assay (Supplementary Fig. 3h) and colony-formation assay (Fig. 3d, e). These results suggested that G-CSF-induced differentiation may confer the cytotoxicity. In the above experiments, mouse neutrophils and human tumor cells were used. To examine if the cell-killing could be due to species difference, we employed human neutrophils. The human promyelocytic leukemia cell line (HL-60), a model for studying myeloid cell differentiation, can be differentiated into mature neutrophils under a combinational treatment with retinoic acid and G-CSF[38]. This was confirmed by flow cytometry of CD66b, a neutrophilic and eosinophilic granulocyte-specific activation antigen[39] (Supplementary Fig. 3l, m). We found that differentiated HL-60 neutrophils, but not the parental cells, show cytotoxicity against LN229$^{TAZ(4SA)}$ tumor cells in all cell survival assays (Fig. 3c–e and Supplementary Fig. 3h). Overall, the above results supported that neutrophil cytotoxicity associates with more differentiated neutrophils.

**Neutrophil-mediated tumor cell killing is achieved via ferroptosis**. To understand the nature of neutrophil-induced tumor cell death, we employed a panel of small-molecule inhibitors, which specifically suppress certain types of regulated cell death. These inhibitors include z-VAD-FMK (apoptosis), necrostatin-1 (necroptosis), GSK-872 (necroptosis), necrosulfonamide (NSA; necroptosis), ferrostatin-1 (ferroptosis), and N-acetyl-L-cysteine (NAC; ROS-induced cell death). These compounds did not affect viability of LN229$^{TAZ(4SA)}$ tumor cells when they were cultured alone (Supplementary Fig. 4a). Interestingly, of these, only ferrostatin-1 and NAC seem to rescue tumor cells from killing by differentiated 32Dcl3 neutrophils (Fig. 4a). Of note, although necrostatin-1 could partially rescue neutrophil-mediated killing, the other two necroptosis inhibitors, GSK-872 and NSA, did not show such an effect (Fig. 4a). Given that necrostatin-1 has been reported to non-specifically inhibit ferroptosis as an off-target effect[40], necroptosis likely was not involved in the cell death we observed. Therefore, this screen suggested that ferroptosis is likely the underlying mechanism of neutrophil-induced tumor cell death. We next confirmed that ferrostatin-1 also suppresses the tumor cell death induced by TANs or differentiated HL-60 neutrophils (Fig. 4b and Supplementary Fig. 4b). Liproxstatin-1 is another ferroptosis inhibitor that is likely a radical-trapping antioxidant[40,41]. We found that liproxstatin-1 similarly rescues tumor cells from killing by the two aforementioned types of neutrophils (Supplementary Fig. 4b, c). To investigate whether such cell death is iron-dependent, we attempted to rescue neutrophilic killing using an iron chelator, deferoxamine (DFO). Indeed, DFO abolished neutrophil-induced tumor cell death (Fig. 4b and Supplementary Fig. 4b). Additionally, we used the Sytox dye assay and further confirmed that neutrophil-induced tumor cell death can be inhibited by ferrostatin-1, liproxstatin-1, or DFO (Supplementary Fig. 4d). Although the exact mechanisms leading to eventual cellular demise in ferroptosis remain to be

defined, it is well-established that ferroptosis is characterized by excessive intracellular accumulation of lipid peroxides. We pondered whether TANs or differentiated neutrophils induce lipid peroxidation in tumor cells, so we subjected neutrophil-tumor cell cocultures to both a general ROS indicator, DCFDA, and a lipid ROS-specific indicator, BODIPY. Tumor cells cocultured with TANs and differentiated 32Dcl3 cells showed greater DCFDA and BODIPY signals than those cultured alone (Fig. 4c, d and Supplementary Fig. 4e, f), revealing higher amounts of lipid peroxidation in the cocultured tumor cells. When ferrostatin-1, liproxstatin-1, or DFO was added into the coculture, accumulation of lipid peroxides, as indicated by BODIPY, was abolished (Supplementary Fig. 4g, h). Ferroptotic cells have been reported to contain smaller and condensed mitochondria[18], and LN229$^{TAZ(4SA)}$ cells cocultured with the cytotoxic differentiated 32Dcl3 and HL-60 cells exhibited such mitochondrial pathology (Fig. 4e), consistent with ferroptotic cell death. Interestingly, mitochondria in these tumor cells also appeared morphologically distorted, with fewer and swollen cristae (Fig. 4e).

Next, to decipher if ferroptosis and thereby lipid peroxide accumulation is associated with tumor necrosis in vivo, we conducted in situ loading of a fluorescent probe detecting intracellular lipid ROS, Liperfluo[42], on unfixed brain slices obtained from LN229$^{TAZ(4SA)}$ tumor-bearing mice. Tumor areas showed stronger Liperfluo signals than surrounding normal brain parenchyma (Fig. 4f, outlined area vs. area marked by blue asterisk). In tumor areas, Liperfluo signals appeared variable. DAPI staining allowed us to distinguish necrotic (DAPI-sparse) from cellular (DAPI-dense) tumor regions (Fig. 4g, left). It appears that necrotic regions had stronger Liperfluo signals than cellular tumor regions (Fig. 4g, red vs. white asterisks). Interestingly, peri-necrotic zones appeared to exhibit the strongest Liperfluo signals (Fig. 4g, white arrows). As such areas are neutrophil-rich (Fig. 2a), we wondered whether it was the neutrophils or tumor cells that contributed to the Liperfluo signals. Using CD45 as a murine leukocyte marker to label singly dissociated cells from LN229$^{TAZ(4SA)}$ tumors, we distinguished immune cells from tumor cells (Supplementary Fig. 4i). It appeared that tumor cells (CD45$^-$) showed stronger Liperfluo signals than CD45$^+$ immune cells (Fig. 4h and Supplementary Fig. 4j). These results indicated that tumor cells in peri-necrotic zones contained higher levels of intracellular lipid peroxides than other tumor cells or non-tumor cells. Expression of PTGS2 (a.k.a. COX2) increases when cells undergo ferroptosis, so it has been commonly used as an indicator for ferroptosis[19,43]. Consistent with our expectations, peri-necrotic tumor cells exhibited increased PTGS2 expression (Fig. 4i). We then characterized mitochondrial morphology in these necrotic tumor cells via electron microscopy of whole-brain sections obtained from LN229$^{TAZ(4SA)}$ tumor-bearing mice. We found that, similar to our in vitro observations with tumor-neutrophil coculture, these necrotic tumor cells contained mitochondria with fewer and seemingly swollen cristae when compared with healthy tumor cells (Fig. 4j). These observations suggested that tumor cells were subject to oxidative stress and accumulated lipid peroxides similar to those cells cocultured with neutrophils in vitro. Collectively, the above results demonstrated that necrotic tumor cells undergo ferroptotic cell death when encountering neutrophils.

**Intercellular transfer of neutrophilic myeloperoxidase (MPO) mediates neutrophil-induced tumor cell cytotoxicity**. To further investigate the neutrophil-tumor interaction in vitro, we used a lipophilic membrane fluorescent dye, PKH26, to label neutrophils and expressed GFP in LN229$^{TAZ(4SA)}$ tumor cells to distinguish

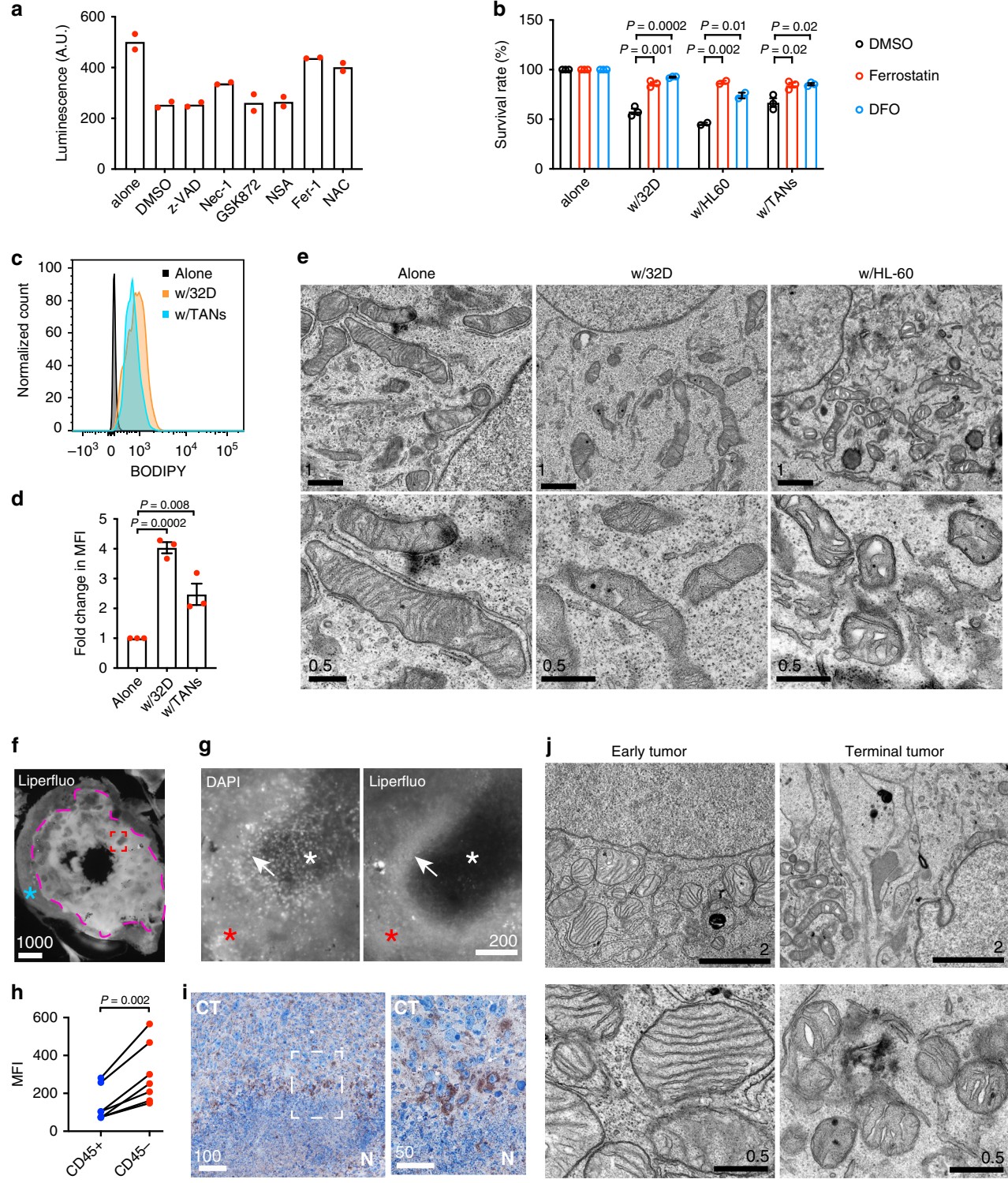

between the two cell types. We noticed that tumor cells contain PKH26+ puncta (Supplementary Fig. 5a). To examine whether these PKH26+ puncta were associated with neutrophil-induced tumor cell death, we quantified the puncta in tumor cells when cocultured with a variety of PKH26-labeled cells. Compared to tumor cells cocultured with tumor-associated or in vitro differentiated murine and human neutrophils, fewer PKH26+ puncta were observed in tumor cells cocultured with tumor-naive neutrophils or undifferentiated myeloid cells, including neutrophils derived from bone marrow and spleen, as well as parental HL-60

and 32Dcl3 cells (Fig. 5a–c and Supplementary Fig. 5b, c). Since the tumor-naive and undifferentiated neutrophils failed to induce tumor cell death (Fig. 3c–e), we speculated that incorporation of PKH26+ puncta by tumor cells may contribute to neutrophils' killing abilities. To test this notion, we separated tumor cells containing more PKH26+ puncta (PKH26high) from those containing fewer or no PKH26+ puncta (PKH26low/-) by flow cytometry (Supplementary Fig. 5d) and compared their survival. We found that PKH26high tumor cells exhibited reduced viability as indicated by the colony-formation assay (Fig. 5d, e) and the Sytox

**Fig. 4 Neutrophil-mediated tumor cell killing is achieved via ferroptosis. a** Viability (luminescence; A.U, arbitrary units) of LN229$^{TAZ(4SA)}$ cells cultured alone or with differentiated 32Dcl3 cells treated with various cell death inhibitors ($n = 2$). **b** Viability (luminescence) of LN229$^{TAZ(4SA)}$ cells cultured alone, with differentiated 32Dcl3, with differentiated HL-60 cells, or with TANs treated with 2 μM ferrostatin-1, 0.1 (w/32D) or 0.2 (w/HL-60 or TANs) mM deferoxamine (DFO), or DMSO. Percent survival is normalized to LN229$^{TAZ(4SA)}$ tumor cells cultured alone and treated with the same compound ($n = 3$). Unpaired two-tailed $t$-test. **c** Representative flow cytometry analysis of BODIPY in LN229$^{TAZ(4SA)}$ cells cultured alone, with TANs, or with differentiated 32Dcl3 cells ($n = 3$). **d** Fold change in median fluorescence intensity (MFI) of flow cytometry analysis in **c** ($n = 3$). Ordinary one-way ANOVA. **e** Low- and high-magnification transmission electron microscopy (TEM) images of LN229$^{TAZ(4SA)}$ tumor cells cultured alone, with differentiated 32Dcl3, or with differentiated HL-60 cells. Five image fields were examined for each condition from two independent experiments with consistent observations. **f** Representative image of in situ Liperfluo dye-loaded brain slice from a LN229$^{TAZ(4SA)}$ tumor-bearing mouse. **g** Representative high-magnification images of the rectangular area outlined in red in **f** showing Liperfluo and DAPI signals. **h** Fold change in MFI of flow cytometry analysis in Supplementary Fig. 4j. CD45$^-$ tumor and CD45$^+$ immune cells were isolated from LN229$^{TAZ(4SA)}$ tumor-bearing mice ($n = 7$). Each pair represents cells isolated from the same animal. Paired two-tailed $t$-test. **i** Chromogenic immunodetection of PTGS2 in a LN229$^{TAZ(4SA)}$ tumor section. The outlined area is enlarged and shown on the right. **j** Representative low- and high-magnification TEM images from LN229$^{TAZ(4SA)}$ tumor-bearing mice brain sections at early tumor progression (left; 16 days after implantation) and upon reaching endpoints (right; 30 days after implantation). For **f**, **g**, **i**, and **j**, three animals were examined independently with similar observations. **a–j** $n$ indicates number of animals or biologically independent samples (with each sample from an independent experiment). Numerical data are presented as mean ± s.e.m. All scale bars are in μm. Source data are provided as a Source Data file.

dye assay (Supplementary Fig. 5e). The reduced viability of PKH26$^{high}$ tumor cells can be rescued by ferrostatin-1, liprox-statin-1, or DFO (Supplementary Fig. 5e).

The correlation between intracellular PKH26$^+$ puncta incorporation and tumor cell death led us to hypothesize that intercellular transfer of neutrophil-specific contents into tumor cells may induce ferroptosis in tumor cells. MPO is a peroxidase mostly expressed in neutrophils, converting hydrogen peroxide to an even more reactive ROS, hydrochlorous acid. We found that the PKH26$^+$ puncta in tumor cells largely co-localized with MPO when cells were stained with an MPO antibody (Fig. 5a and Supplementary Fig. 5b). Similar to the PKH26$^+$ puncta, MPO$^+$ puncta were also more frequently found in tumor cells cocultured with cytotoxic neutrophils (i.e., TANs, differentiated HL-60 and 32Dcl3 cells; Fig. 5a, c and Supplementary Fig. 5b, c). To understand whether such neutrophil-derived MPO transfer into tumor cells required direct cell–cell contact, we employed two approaches. First, tumor cells were cultured in conditioned media (CM) derived from neutrophil monoculture. Second, tumor cells and neutrophils were cocultured with a cell-impermeable membrane in between, with tumor cells on the bottom of the wells and neutrophils above the membrane. In both conditions, barely any MPO$^+$ puncta were observed in tumor cells when compared to those cocultured with neutrophils directly (Supplementary Fig. 5f, g). These results together suggested that direct cell–cell contact or close proximity between tumor cells and neutrophils is necessary for intercellular MPO transfer.

We further examined whether the MPO puncta also exist in vivo via immunohistochemistry. In a LN229$^{TAZ(4SA)}$ tumor section, the MPO-expressing cells are mostly neutrophils in the necrotic area (Fig. 5f, green-outlined). These cells are round and smaller. MPO-staining signals in these cells cannot be clearly separated from the nuclei, presumably due to relatively condensed cytoplasm (Fig. 5f, arrows in the magnified green-outlined area). Interestingly, some peri-necrotic tumor cells appear to contain MPO puncta (Fig. 5f, magenta-outlined). These cells have large nuclei and irregular shapes. MPO-staining signals in these cells are punctate and can be clearly separated from the nuclei (Fig. 5f, arrowhead in the magnified magenta-outlined area). To further confirm this observation, we conducted a double staining of MPO and N-cadherin, which is highly expressed in GBM cells. N-cadherin also allowed us to examine individual cells in tissue. MPO$^+$ puncta can be seen in tumor cells (Fig. 5g, yellow arrowheads). Notably, an infiltrated mouse neutrophil containing more, but smaller, MPO$^+$ puncta was also present here (Fig. 5g, white arrows). Similar to the above

observation (Fig. 5f), MPO-staining signals in this putative neutrophil appeared to overlap with the nucleus in the merged image. This cell can be distinguished from surrounding human tumor cells by characteristic bright and condensed DAPI-stained chromocenters commonly found in mouse cells but not human tumor cell lines. Overall, these results suggested that tumor cells are able to obtain MPO from neutrophils both in vitro and in vivo. We then tested if MPO enzymatic activity is necessary for neutrophil-induced tumor cell death. 4-aminobenzoic acid hydra-zide (4-ABAH) and PF06282999 are two small-molecule inhibitors, which can specifically inhibit the peroxidase activity of MPO[44,45]. Addition of 4-ABAH into neutrophil-tumor cell cocultures abolished tumor cell death (Fig. 5h). Similarly, PF06282999 also inhibited neutrophil-induced tumor cell death (Fig. 5h). These results suggested that MPO enzymatic activity is involved in the neutrophil cytotoxicity. To further examine this notion, *MPO* was knocked down from differentiated HL-60 and 32Dcl3 cells by two different short-hairpin RNA (shRNAs; Supplementary Fig. 5h, i). These MPO-depleted cells were significantly less capable of inducing tumor cell death than controls when examining via Sytox dye viability assay (Fig. 5i), concordant with results from pharmacological inhibition (Fig. 5h). Taken together, the above results support the notion that neutrophilic granules are transferred intercellularly into tumor cells, and it is the neutrophilic granular contents, such as MPO as demonstrated here, which induce ferroptosis in tumor cells.

**GPX4 overexpression or ACSL4 depletion in tumor cells dampens necrosis and tumor aggressiveness.** To assess whether ferroptosis is responsible for tumor necrosis, we evaluated the impact of ferroptosis blockade on necrosis. GPX4 is the essential phospholipid peroxidase responsible for reducing lipid per-oxides, thereby protecting cells from ferroptosis[43]. We found that the amount of GPX4 is less in tumor cells close to the necrotic area than in the cellular tumor region in LN229$^{TAZ(4SA)}$ tumors (Fig. 6a). This observation suggested that GPX4 deple-tion may predispose tumor cells to ferroptosis. To examine if GPX4 replenishment could rescue the tumor cells from fer-roptosis, we ectopically expressed recombinant GPX4 (rGPX4) in LN229$^{TAZ(4SA)}$ cells (Supplementary Fig. 6a). Of note, rGPX4 (arrow) contains a mitochondria-targeting sequence[46] and appears to be larger than endogenous GPX4 (asterisk), which likely does not have this sequence. Ectopically expressed rGPX4 appeared to be diffuse throughout the cytosol and not limited to the mitochondria (Supplementary Fig. 6b). LN229$^{TAZ(4SA)}$ cells expressing rGPX4 are less sensitive to neutrophil-mediated killing than control cells in the in vitro coculture assay

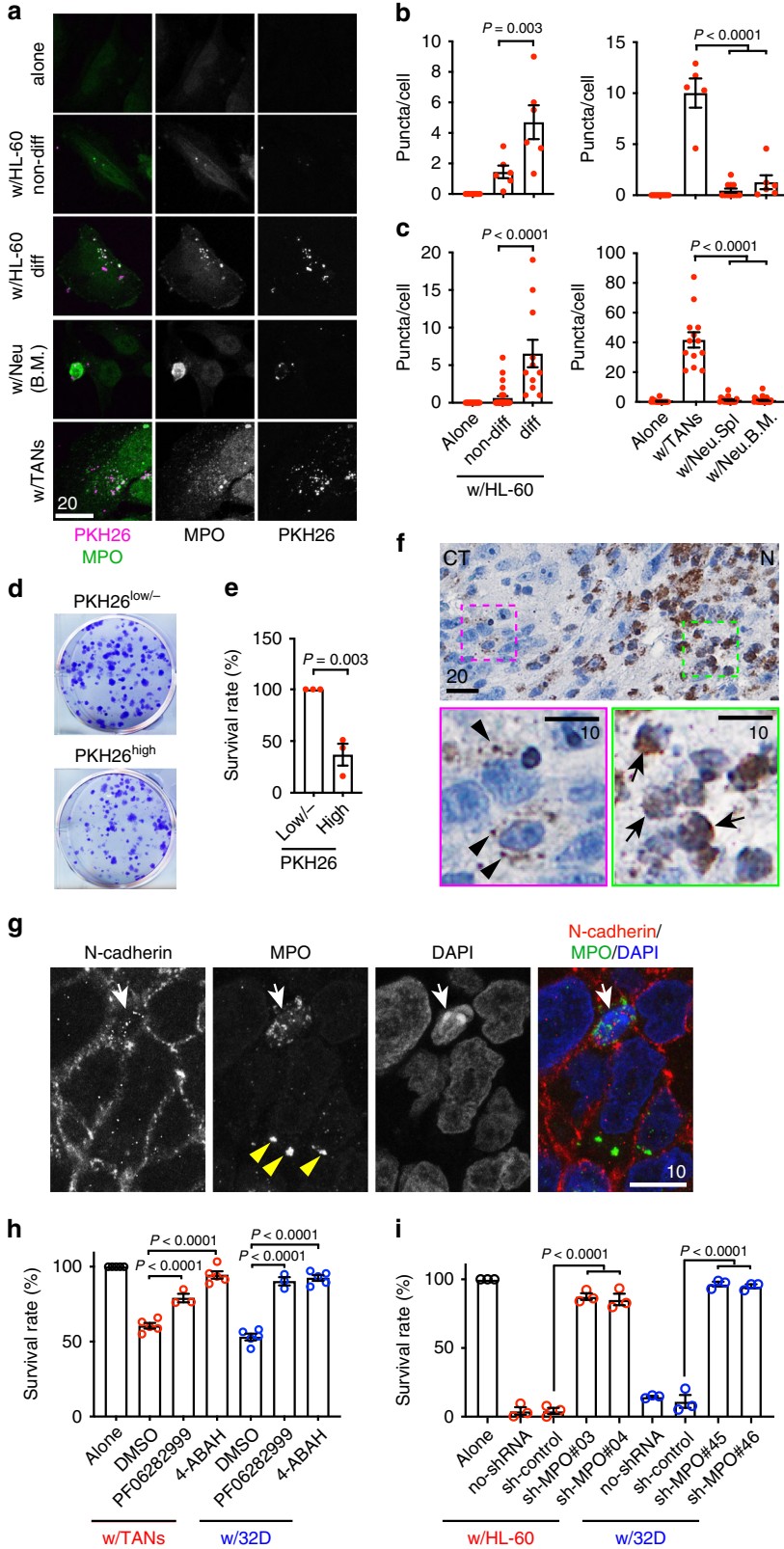

(Supplementary Fig. 6c). Mice bearing rGPX4-expressing LN229$^{TAZ(4SA)}$ tumors lived significantly longer (24%) than the controls (Fig. 6b). Histological evaluation of tumors showing similar sizes from the two groups indicated that the necrosis in rGPX4-expressing tumors was significantly smaller (47%) than in the control tumors (Fig. 6c and Supplementary Fig. 6d). Therefore, the above results supported the notion that inhibiting ferroptosis lessens tumor necrosis and aggressiveness. Interestingly, expression of rGPX4 slightly slowed tumor growth, although the difference was statistically insignificant

**Fig. 5 Intercellular transfer of neutrophilic myeloperoxidase (MPO) is responsible for neutrophil-mediated tumor cell cytotoxicity. a** MPO immunofluorescent staining of LN229$^{TAZ(4SA)}$ cells cultured alone or with indicated PKH26-labeled neutrophils. Non-diff undifferentiated, diff differentiated, Neu neutrophils, B.M. bone marrow. **b, c** PKH26 (**b**) and MPO (**c**) puncta per cell in **a**. Each data point represents an average of 10–15 cells in one image field. In **b**, $n_{alone} = 10$, $n_{non-diff} = 6$, $n_{diff} = 6$, $n_{w/TANs} = 5$, $n_{Neu.Spl} = 11$, and $n_{Neu.B.M.} = 6$. In **c**, $n_{alone} = 16$, $n_{non-diff} = 30$, $n_{diff} = 11$, $n_{w/TANs} = 13$, $n_{Neu.Spl} = 27$, and $n_{Neu.B.M.} = 18$. $n$ indicates number of image fields from three independent experiments or animals. Ordinary one-way ANOVA. **d** Images of colony-formation assays of ZsGreen-expressing LN229$^{TAZ(4SA)}$ cells with many (PKH$^{high}$) or few (PKH$^{low/-}$) PKH26 puncta when cocultured with PKH26-labeled differentiated 32Dcl3 cells. **e** Percent survival of conditions in **d** normalized to PKH$^{low/-}$ LN229$^{TAZ(4SA)}$ cells. $n = 3$. Unpaired two-tailed $t$-test. **f** Chromogenic immunodetection of MPO in an endpoint LN229$^{TAZ(4SA)}$ tumor section. The outlined areas are shown on the bottom. **g** Immunofluorescent staining on an endpoint LN229$^{TAZ(4SA)}$ tumor section. For **f** and **g**, three animals were examined independently with similar observations. **h** Viability (luminescence) of LN229$^{TAZ(4SA)}$ cells cultured alone ($n = 5$) or with indicated neutrophils, treated with either of two MPO inhibitors, 4-ABAH (2 μM; $n_{TANs} = 5$ and $n_{32Dcl3} = 5$) or PF06282999 (2 μM; $n_{TANs} = 3$ and $n_{32Dcl3} = 3$), or with DMSO ($n_{TANs} = 5$ and $n_{32Dcl3} = 5$). Percent survival normalized to LN229$^{TAZ(4SA)}$ cells cultured alone and treated with the same compound. Ordinary one-way ANOVA. **i** Viability (evaluated by Sytox-Blue) of LN229$^{TAZ(4SA)}$ cells cultured alone or with indicated neutrophils non-transduced (no-shRNA) or transduced with scrambled shRNA (sh-control) or shRNAs targeting *MPO*. $n = 3$. For **e**, **h**, and **i**, $n$ indicates number of independent experiments. Each data point represents an animal or average of replicates from an independent experiment. Ordinary one-way ANOVA. **a–i** Numerical data presented as mean ± s.e.m. Scale bars in μm. Source data provided as a Source Data file.

(Supplementary Fig. 6e). This suggested that reduced tumor growth may not wholly explain prolonged survival.

To further investigate these notions, we employed another approach to inhibit ferroptosis. ACSL4 is an essential enzyme that synthesizes polyunsaturated fatty acid-containing phospholipids, the primary substrates for lipid peroxidation and subsequent ferroptosis[47,48]. To inhibit ferroptosis, we silenced the expression of *ACSL4* in LN229$^{TAZ(4SA)}$ cells with two different shRNAs (Supplementary Fig. 6f, g). Tumor cell-killing by TANs or in vitro differentiated neutrophils was dampened when *ACSL4* expression was more effectively silenced (Supplementary Fig. 6h, sh#41). However, when ACSL4 was partially depleted, inhibition of cell-killing was not significant (Supplementary Fig. 6h, sh#42). Histological evaluation of similar-sized tumors from mice implanted with ACSL4-depleted LN229$^{TAZ(4SA)}$ cells indicated that the necrosis in ACSL4-depleted tumors was significantly (68% and 58%, respectively) smaller than in the control tumors (Fig. 6d and Supplementary Fig. 6i). Mice implanted with tumor cells knocked down of *ACSL4* by either shRNA lived significantly longer (45% and 31%, for sh#41 and sh#42, respectively) than controls (Fig. 6e). Notably, tumor growth was dampened when *ACSL4* expression was more effectively silenced (Supplementary Fig. 6j, sh#41). However, when ACSL4 was partially depleted, there was little inhibition of tumor growth (Supplementary Fig. 6j, sh#42). Results from the two above approaches supported that ferroptosis is involved in the formation of tumor necrosis, and together they suggested that ferroptosis enhances tumor aggressiveness.

To understand the mechanism of the pro-tumorigenic role of ferroptosis-mediated cell death, we speculated that the dying tumor cells may secrete certain factors, which promote tumor progression. To examine this, we carried out an unbiased screen for these factors via a human cytokine antibody array using whole-tumor lysates. Five factors appeared to be more abundant in LN229$^{TAZ(4SA)}$ (with necrosis) than LN229$^{vector}$ (without necrosis) tumors (Fig. 6f). We then conducted comparative studies to examine whether these factors were associated with necrosis in human GBMs. To this end, we analyzed gene expression using a publicly available GBM dataset (Ivy Glioblastoma Atlas Project)[49]. This dataset comprises 26 samples from the tumor peri-necrotic zone (PNZ) and 111 samples from the cellular tumor zone (CT). In this analysis, we found that four of these five genes show increased expression in the peri-necrotic zone compared to the cellular tumor region of human GBMs (Fig. 6g). Ingenuity pathway analysis (IPA) of genes differentially expressed in these two distinct geographical features suggested that these four genes are more activated in the peri-necrotic area

(Fig. 6g). Since the *IL8* data are not available in the TCGA dataset, we further examined the other three genes in the TCGA dataset. They show higher expression in the GBMs of mesenchymal subtype than in GBMs of the classical or proneural subtypes (Fig. 6h) and are associated with poorer survival (Fig. 6i). Notably, *IL8* and *IL6* have both been implicated in promoting GBM progression[50]. Furthermore, we examined the impact on the GBM microenvironment when ferroptosis is inhibited by the above two genetic manipulations. It appeared that numbers of CD31$^+$ endothelial cells and GFAP$^+$ astrocytes were not affected (Supplementary Fig. 6k, l). Interestingly, when examined by TMEM119, a murine microglia marker, we saw fewer intratumoral microglia in tumors expressing rGPX4 or *ACSL4* shRNAs. In contrast, brain parenchymal microglia showed no difference (Supplementary Fig. 6k, l). Overall, these results suggested that secretion of the pro-tumorigenic factors and/or recruiting microglia by ferroptosed tumor cells allows ferroptosis to play a pro-tumorigenic role and contribute to tumor aggressiveness and poor outcomes in human GBMs.

**Ferroptosis is associated with mesenchymal transition and positively correlated with tumor aggressiveness in human GBMs.** To explore if ferroptosis is associated with human GBM necrosis, we generated a list of ferroptosis regulatory genes (Supplementary Fig. 7a). These genes were reported to be involved in ferroptosis and can be classified as either ferroptosis-promoting or ferroptosis-inhibiting genes[19,20]. Gene set enrichment analysis (GSEA)[51] of the TCGA GBM dataset found that ferroptosis-promoting genes are more upregulated in mesenchymal (MES) GBM than in the proneural (PN) subtype (Fig. 7a). Since MES-GBM is associated with shorter survival and is more likely to have necrosis[52–54], this result suggests an association between ferroptosis and necrosis, as well as poor prognoses. As traditional tumor subtyping of a whole tumor may not reflect intratumoral heterogeneity[55], to examine the association between ferroptosis and necrosis more directly, we analyzed the aforementioned Ivy GBM dataset. GSEA found that ferroptosis-promoting genes are more upregulated in the PNZ, while ferroptosis-inhibiting genes have no enrichment in either tumor region (Fig. 7b). These results suggested that ferroptosis takes place specifically in the PNZ. Comparison of the most enriched ferroptosis genes in MES-GBMs and in the PNZ of GBMs revealed substantial overlap (Fig. 7c), suggesting a link between the MES subtype and the PNZ. To further examine ferroptosis in human GBMs, we used PTGS2 as a marker. Similar to our observation in tumor-bearing mice (Fig. 4i), PNZ tumor cells showed much stronger expression of PTGS2 when examined via

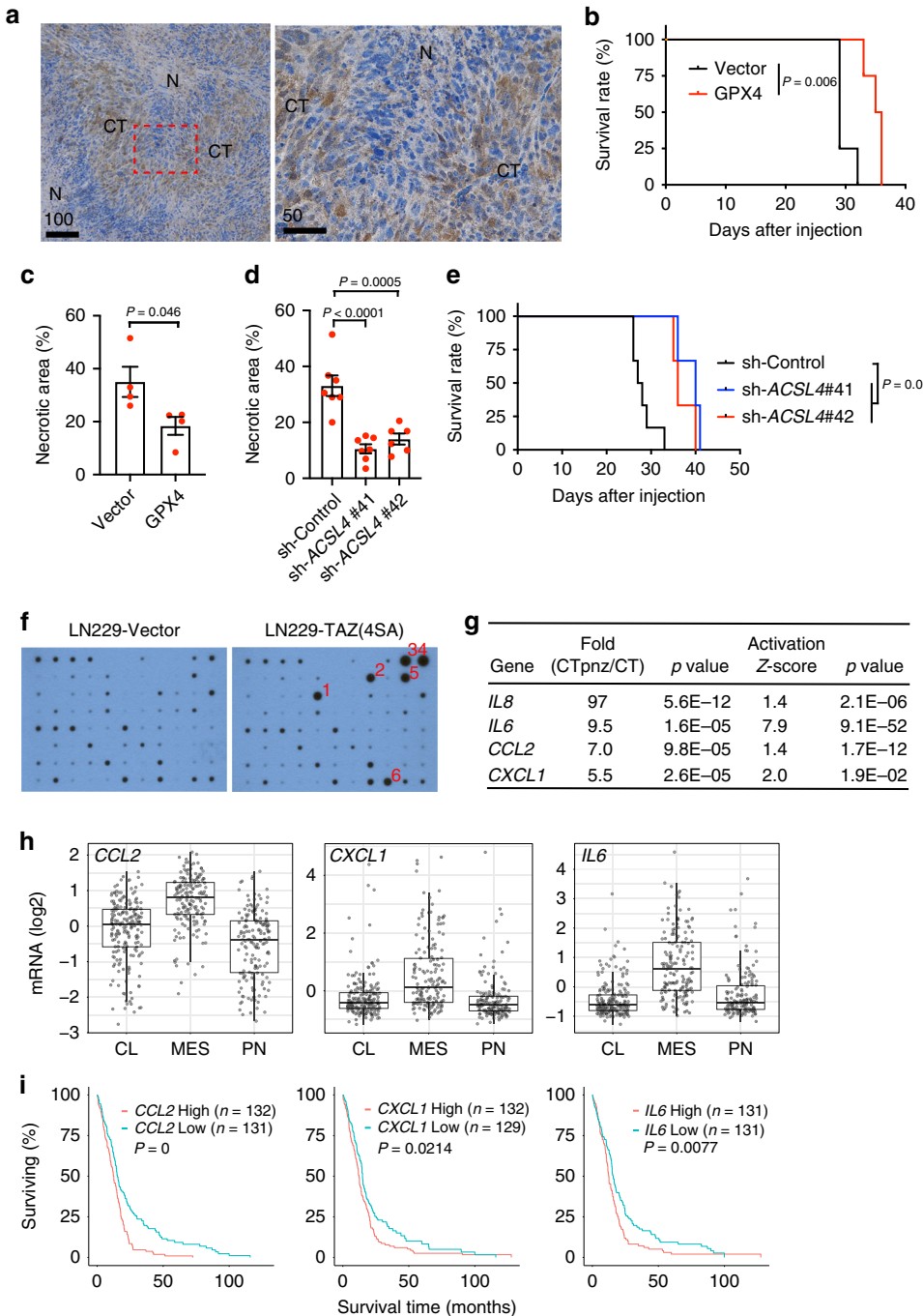

**Fig. 6 GPX4 expression and ACSL4 depletion inhibit tumor necrosis and progression. a** Chromogenic immunodetection of GPX4 in an endpoint LN229$^{TAZ(4SA)}$ tumor section. N necrotic region, CT cellular tumor region. The outlined area is shown on the right. Three animals were examined independently with similar observations. **b** Kaplan–Meier survival curve of mice implanted with LN229$^{TAZ(4SA)}$ cells transduced with vector or rGPX4. $n = $ 4 mice for each group. Two-sided log-rank test. **c** Necrosis-to-tumor area ratios of tumors as indicated in **b** upon reaching same tumor size as measured by bioluminescence imaging. $n = 4$. Unpaired two-tailed $t$-test. **d** Necrosis-to-tumor area ratios of tumors developed from LN229$^{TAZ(4SA)}$ cells transduced with scrambled shRNA (sh-Control; $n = 7$) or shRNAs targeting ACSL4 (sh #41 $n = 7$; sh #42 $n = 6$). Ordinary one-way ANOVA. In **c** and **d**, $n$ indicates number of tumors. Each data point represents an animal. Data presented as mean ± s.e.m. **e** Kaplan–Meier survival curve of mice implanted with tumors indicated in **d**. $n_{\text{sh-Control}} = 6$, $n_{\text{sh-ACSL4 #41}} = 3$, and $n_{\text{sh-ACSL4 #42}} = 3$. Two-sided log-rank test. $n$ indicates number of mice. **f** Cytokine arrays of indicated tumor lysates. (1) CCL2, (2) IL6, (3) CXCL1, (4) CXCL family (CXCL1, CXCL2, and CXCL3), (5) IL8, and (6) TIMP2. **g** Differential expression (analyzed through https://glioblastoma.alleninstitute.org) and predicted activation (analyzed through IPA; right-tailed Fisher's exact test) of indicated genes. **h** Expression of indicated genes in GBM subtypes. Data presented as box-and-whisker plots with boxes marking first quartile, median, and third quartile and with whiskers extending 1.5 times the inter-quartile range. Each point represents a tumor ($n_{CL} = 199$; $n_{MES} = 166$; $n_{PN} = 163$). CCL2: $P_{\text{MES-PN}} = 1.7\text{e-32}$; $P_{\text{MES-CL}} = 2.7\text{e-18}$. CXCL1: $P_{\text{MES-PN}} = 4.9\text{e-15}$; $P_{\text{MES-CL}} = 8.5\text{e-14}$. IL6: $P_{\text{MES-PN}} = 1.4\text{e-22}$; $P_{\text{MES-CL}} = 4.4\text{e-33}$. Pairwise $t$-tests with Bonferroni correction. **i** Survival of GBM patients showing higher (top 25%) or lower (bottom 25%) expression of indicated genes. Log-rank test. $n$ indicates number of human subjects. In **h** and **i**, analysis of TCGA dataset through GlioVis. Scale bars in μm. Source data provided as a Source Data file.

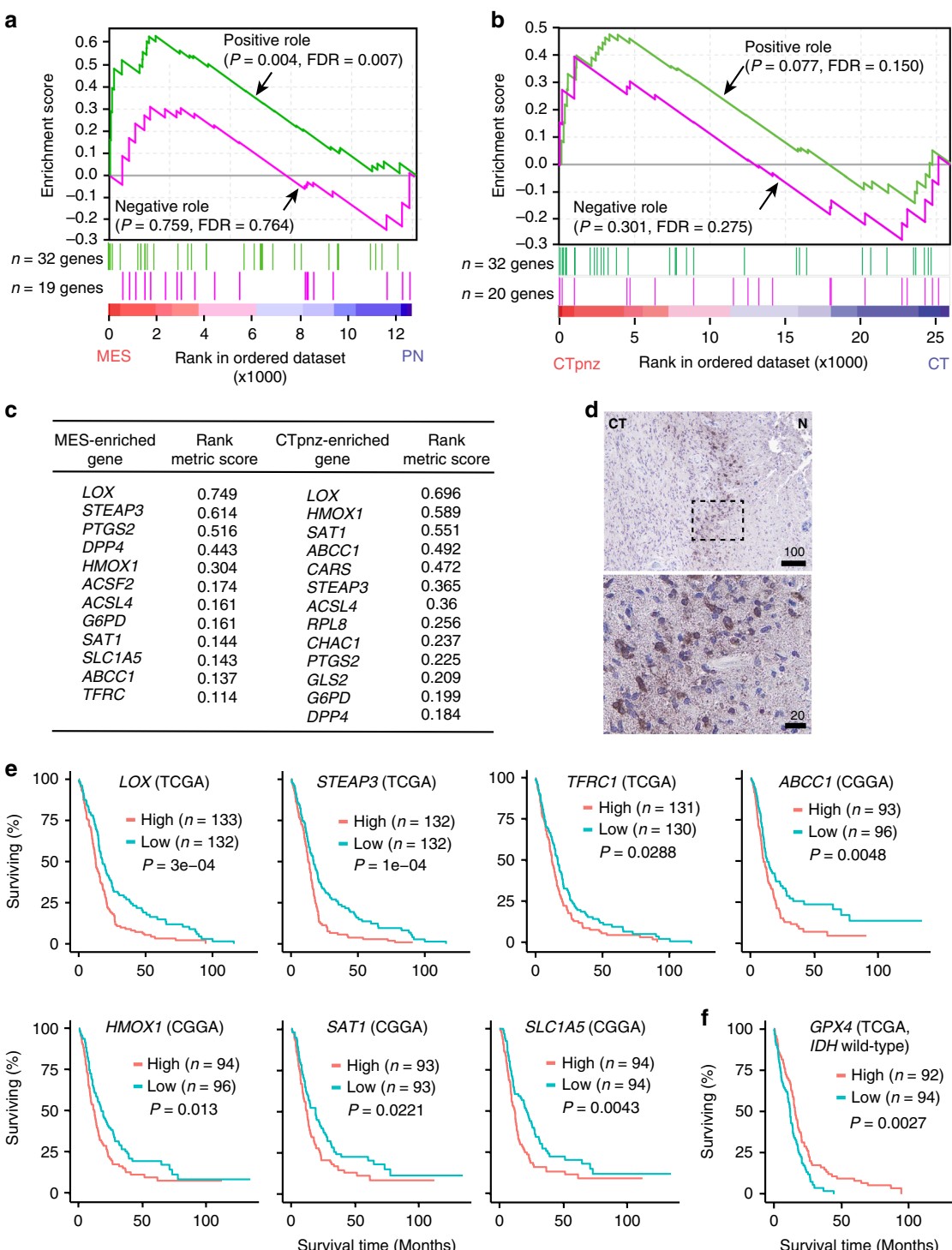

**Fig. 7 Ferroptosis is associated with mesenchymal transition and shorter survivals in human GBMs. a** Gene set enrichment analyses (GSEA) of ferroptosis-related genes as listed in Supplementary Fig. 7a using the TCGA GBM dataset; proneural (PN; $n = 166$ tumors) and mesenchymal (MES; $n = 170$ tumors). Nominal $P$-values and the false-discovery rate (FDR)[51] are indicated. **b** GSEA of ferroptosis-related genes in GBM gene expression dataset obtained from Ivy GBM Atlas comparing cellular tumor peri-necrotic zone (denoted by CTpnz) and cellular tumor zone (denoted by CT). Nominal $P$-values and the false-discovery rate (FDR)[51] are indicated. ($n = 26$ tumor samples for CTpnz; $n = 111$ tumor samples for CT). **c** Comparison of most enriched ferroptosis genes in the MES-GBM subtype and the CTpnz in GBM shows a marked overlap. **d** Representative image of chromogenic immunodetection of PTGS2 (a.k.a. COX2) in a formalin-fixed paraffin-embedded, human GBM brain section. The outlined areas are enlarged and shown on the bottom. Specimens from three different patients were examined independently with similar observations. **e** Survival analyses of GBM patients showing higher (top 25%) or lower (bottom 25%) expression of indicated genes. (TCGA or CGGA dataset was analyzed through GlioVis.) Log-rank $P$-value of each graph is shown. $n$ indicates total number human subjects. **f** Survival analyses of GBM patients showing higher (top 25%) or lower (bottom 25%) expression of *GPX4*. (The *IDH1* wild-type cohort TCGA dataset was analyzed through GlioVis.) Log-rank $P$-value is shown. $n$ indicates total number human subjects. All scale bars are in μm.

immunohistochemical staining (Fig. 7d), indicating the occurrence of ferroptosis in this area. In further study of the association between ferroptosis and tumor aggressiveness, we found that higher expression of the two most enriched ferroptosis genes, lysyl oxidase (*LOX*) and six-transmembrane epithelial antigen of prostate 3 (*STEAP3*), as well as five other highly enriched ferroptosis genes, are all associated with poorer survival in GBM patients (Fig. 7e). Notably, although *GPX4* does not show an effect on survival in the patient cohort with all the TCGA GBM cases, lower expression of *GPX4* is associated with poorer survival in the cohort without *IDH1* mutations (Fig. 7f). This result suggests that *GPX4* expression does impact tumor aggressiveness negatively, and such a relationship may depend on the metabolic status of tumors. Overall, these results further support that ferroptosis is associated with more aggressive tumors.

**TANs positively correlate with the extent of tumor necrosis and predict poor survival in GBM patients**. To explore if neutrophils are associated with human GBM necrosis, we conducted ingenuity pathway analysis (IPA) of genes differentially expressed in the PNZ and CT samples of the Ivy GBM dataset. The results showed that cellular processes involving neutrophil recruitment, movement, accumulation, and degranulation are all upregulated in the PNZ when compared to CT (Supplementary Fig. 8a), suggesting neutrophils are more active in the PNZ. In addition, cellular functions related to ferroptosis, such as ROS generation, lipid metabolism, and synthesis of polyunsaturated fatty acids (PUFAs), are also enhanced in the PNZ (Supplementary Fig. 8a). This result is in agreement with the above analysis showing ferroptosis is associated with human GBM necrosis. Moreover, the PNZ is also associated with enhanced cellular processes related to advanced malignant solid tumors (Supplementary Fig. 8a). To further elucidate the interconnection between GBM necrosis and neutrophils, we conducted volumetric analyses of tumors and tumor necrosis using pre-operative, T1-weighted post-contrast magnetic resonance imaging (MRI) studies collected from a cohort of GBM patients (Fig. 8a). Tumor size and tumor necrosis correlated with each other (Supplementary Fig. 8b). To examine if development of tumor necrosis directly correlated with circulating neutrophils, we obtained pre-operative neutrophil counts from patients' complete blood counts (CBC) via electronic medical chart searches. While the number of circulating neutrophils, as measured in percentage of total white blood cells, positively correlated with both tumor and necrotic core sizes (Supplementary Fig. 8c, d), the correlation between neutrophil count and the necrosis-to-tumor ratio was even stronger (Fig. 8b). These results suggested that neutrophil production and mobilization from bone marrow to blood associate with the extent of tumor necrosis. Of note, this study cannot exclude a possible effect on circulating neutrophil counts by infections or medication, such as perioperative steroids. To examine neutrophil infiltration in GBMs, we examined a cohort of 45 GBM samples obtained from deceased patients whose pathology reports specified "prominent" or "extensive" necrosis. Neutrophils in H&E-stained sections can be identified by their characteristic multilobed nuclei (Fig. 8c, arrows). With this approach, we counted the absolute number of neutrophils per high-power field (40x objective) in three distinct regions of interest, including tumor-infiltrating, necrotic, and interface zones. Cells with the characteristic appearance of neutrophils were almost exclusively found in the necrotic zones (Fig. 8d). To identify neutrophils more definitively, we performed immunohistochemistry on tumor sections obtained from this cohort, using CD66b, a neutrophilic and eosinophilic granulocyte-specific activation antigen[39]. Immunohistochemical staining verified the cells quantified

in H&E staining were indeed neutrophils and revealed that CD66b-positive cells localized specifically in the tumor necrotic zones (Fig. 8e). To examine if necrosis and neutrophils are associated with poor survival, we separated the patient cohorts into two groups based on the necrosis-to-tumor volumetric ratios and tumor-neutrophil count. The results showed that both necrosis-to-tumor ratio and higher infiltration of neutrophils correlate with poorer survival (Fig. 8f, g). Above all, these studies supported the notion that, consistent with our pre-clinical model, neutrophil infiltration and ferroptosis are closely associated with tumor necrosis and predict poor survival in GBM patients.

## Discussion

Necrosis is commonly found in advanced solid tumors, especially GBMs. However, the nature and mechanisms driving tumor necrosis remain obscure. In this study, we demonstrated that ferroptosis is involved in the development of necrosis in GBMs. This regulated cell death can be triggered by neutrophils, which are likely recruited into tumors by certain early tissue damage(s) during tumorigenesis. Mechanistically, we showed that activated mature neutrophils induce lipid peroxidation in tumor cells by transferring their characteristic granules into tumor cells. The granules contain myeloperoxidase, which is responsible for increasing tumor cellular lipid-based ROS, thereby leading to the ferroptotic cell death.

While conventional models propose that tumor necrosis occurs when the blood supply is insufficient to support tumor growth, the formation of necrosis appears to be more complex[11]. In this study, tumors derived from vector- or TAZ(4SA)-transduced GBM cells provided unique models to dissect the cell death process in GBM necrosis. LN229$^{TAZ(4SA)}$ tumors develop necrosis as early as day 16 after implantation (Fig. 2g). At this time, tumor sizes are still relatively small. In contrast, LN229$^{vector}$ tumors do not develop necrosis even at sizes that are much larger than LN229$^{TAZ(4SA)}$ tumors (Fig. 2g vs. Supplementary Fig. 1c). This indicates that tumor size does not determine necrosis development, and this is consistent with clinical observations that small GBMs can also develop necrosis[11]. Although the exact instigator of necrosis in LN229$^{TAZ(4SA)}$ tumors remains to be determined, the foci, which appear to be necrotic niches are infiltrated by neutrophils (Fig. 2g). These neutrophils may contribute to necrosis formation during an early-stage of necrosis development (i.e., neo-necrosis). Since neutrophil depletion cannot completely eliminate necrosis formation (Fig. 3a), our results cannot exclude contributions by other factors, such as ischemia, to necrosis induction. While it was reported that the anti-Ly6G antibody that we used can specifically recognize neutrophils but not monocytes/macrophages[31], we cannot exclude a possible off-target effect on the resident immune cells, such as microglia. Therefore, whether microglia play a role in necrosis is an open question. Nevertheless, the results did support that neutrophils play a remarkable role in promoting necrosis development to its fullest extent. How neutrophils are recruited to these foci remains to be resolved. It is unlikely that TAZ activation itself causes neutrophils' infiltration into LN229$^{TAZ(4SA)}$ tumor because no neutrophil accumulation was found in these tumors at day 10 after implantation (Fig. 2g). Hypoxia-induced tumor cell inflammation was proposed to recruit neutrophils in a mouse model of PTEN-deficient uterine cancer[56]. Moreover, in non-neoplastic tissue inflammation, damaged/dying cells are known to passively release DAMPs, such as HMGB1, which preferentially recruit neutrophils over other immune cells[57]. Therefore, it is plausible that initial hypoxic cellular demise or inflammation might be responsible for neutrophil recruitment. The observation that the Ly6G$^+$-enriched

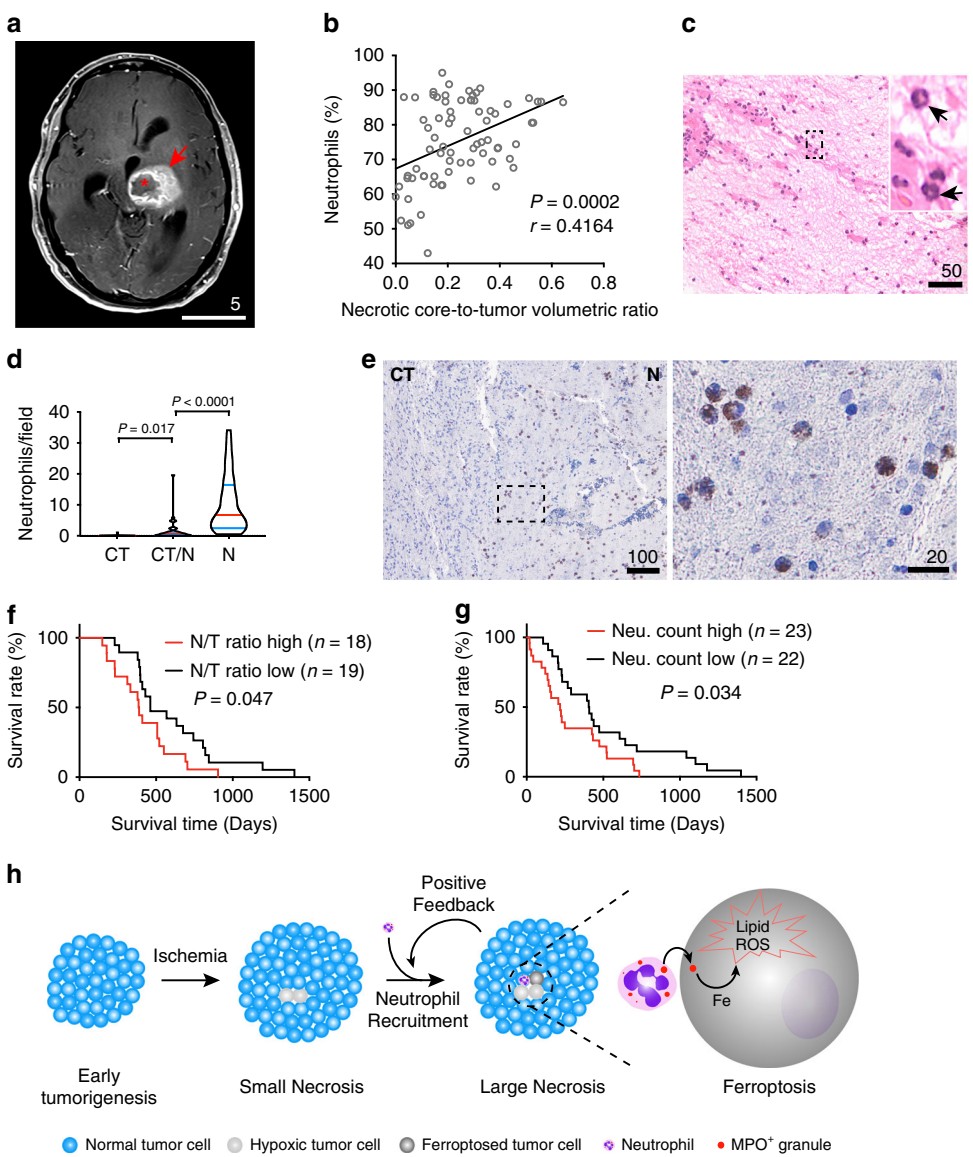

**Fig. 8 Tumor-associated neutrophils, both peripherally and intra-tumorally, are closely associated with tumor necrosis in human GBMs and shorter survivals. a** Example of a pre-operative T1-weighted fat-saturated post-contrast MRI obtained from a GBM patient included in the study cohort as in **b**; necrotic core (asterisk); tumor (arrow). **b** Correlation analyses of pre-operative T1-weighted MRI-identified necrosis-to-tumor volumetric ratios and peripheral neutrophil count reported in blood differentials in a GBM patient cohort ($n = 75$ human subjects), Pearson's correlation coefficient test. **c** Representative H&E-stained formaldehyde-fixed paraffin-embedded human GBM section containing the necrotic zone. A total of 45 patients were examined with similar observations. **d** Comparison of absolute numbers of neutrophils/high-power field among cellular tumor (CT), necrosis (N), and CT-N interface zones (CT/N) in human GBM sections obtained from a deceased patient cohort ($n = 45$ patients); Brown-Forsythe and Welch one-way ANOVA tests. **e** Chromogenic immunodetection of human granulocyte marker, CD66b, in a formaldehyde-fixed paraffin-embedded, human GBM brain section obtained from the study cohort used for quantification in **d**. The outlined areas are enlarged and shown on the right. Specimens from three different patients were examined independently with similar observations. **f** Kaplan–Meier survival curve of the patient cohort shown in **b**; all deceased patients ($n = 37$ patients) from this cohort were included for survival analysis. Patients were separated into two groups by median radiographically identified necrosis-to-tumor (N/T) volumetric ratios; two-sided log-rank test. **g** Kaplan–Meier survival curve of the patient cohort as shown in **d**; all patients from the cohort shown in **d** were included in this survival analysis ($n = 45$ patients). Patients were separated into two groups by median absolute neutrophil count/high-power field in necrotic region obtained from their diagnostic H&E-stained sections. Two-sided log-rank test. **h** Current proposed model of how TANs contribute to augmenting GBM necrosis development to its fullest extent via neutrophil-mediated ferroptotic tumor cell death. All scale bars are in μm, except in Fig. 8a, whose scale bar is in cm. Source data are provided as a Source Data file.

CD11b^high^CD45^high^ cell population progressively outnumbers other myeloid cell populations as tumors develop (Supplementary Fig. 2a, b) suggests that, as necrosis progresses and enlarges, the damaged/dying tumor cells may recruit even more neutrophils to infiltrate into necrotic niches. That is, necrosis and neutrophil infiltration may form a positive feedback loop to amplify intra-tumoral necrosis formation to its fullest capacity (Fig. 8h).

Of note, the mouse models in this study are immunodeficient, so whether neutrophil-induced tumor cell ferroptosis also plays a similar role in tumor necrosis in an immunocompetent system warrants further study.

Why necrosis commonly occurs in GBMs is unknown. Given that ferroptotic cell death is the cause of tumor necrosis, such commonality may be due to the intrinsically deregulated signaling

in combination with the unique tumor microenvironment. It was found that the GBMs of MES subtype have more necrosis than other subtypes[52–54]. TAZ is one of the three transcriptional regulators to drive the gene expression program associated with GBM MES differentiation[25,58]. Interestingly, activation of TAZ or its paralog, Yes-associated protein (YAP), is able to promote ferroptotic cell death under canonical ferroptosis stimuli[59,60]. Therefore, TAZ activation may sensitize GBM cells to ferroptosis. In addition, RAS-RAF-ERK signaling is one of the most commonly activated pathways in GBMs[61]. This signaling was also linked to ferroptosis sensitivity, although the effect seems to be cell-type-specific[43,62–64]. Therefore, it would be interesting to investigate whether RAS-RAF-ERK signaling activation also contributes to ferroptosis in GBMs. Although such deregulated signaling pathways could sensitize GBM cells to ferroptosis, intracellular ROS imbalance is required to trigger cell death. Our finding that neutrophils can initiate tumor cell ferroptosis suggests that these immune cells could be the trigger. In addition, elevated levels of glutamate were observed in necrotic GBMs[65], and it was proposed that accumulation of extracellular glutamate may cause necrosis[66]. Notably, elevation in extracellular glutamate could induce ferroptosis by inhibiting cystine uptake through system Xc-[18]. Therefore, increased levels of glutamate in the GBM microenvironment may also trigger tumor cell ferroptosis.

Although the anti-tumor cytotoxicity of neutrophils has been studied for decades, how tumor cells die under these circumstances is elusive. This study revealed that tumor cells undergo ferroptotic death when encountering neutrophils infiltrating into the tumor stroma. Ferroptosis induction by neutrophils appears to require certain activation signals, likely provided by the tumor microenvironment. Our in vitro studies showed that G-CSF can activate neutrophil-induced tumor cell cytotoxicity. Since GBM cells can synthesize G-CSF[67], this cytokine may be responsible for activating neutrophils in the tumor microenvironment. In addition, we observed in vitro tumor cell-killing capacities by G-CSF-differentiated 32Dcl3 (murine cells) and HL-60 (human cells), while such killing capacity was not seen with neutrophils isolated from the bone marrow of non-tumor-bearing mice. The discrepancy may be due to the differences in their stages of maturation (i.e., differentiation). Although neutrophils have conventionally been considered a homogeneous population with well-defined molecular signatures and largely conserved functions, mounting evidence suggests that neutrophils exist as a heterogeneous population with versatile immunophenotypes and dynamic functional plasticity, mostly thought to be context- and environment-dependent[68,69]. A recent study using mass cytometry (CyTOF) suggested that granulocytes residing in the bone marrow can be divided into at least three distinct subsets (committed proliferative neutrophil precursors, immature neutrophils, and mature neutrophils), each with distinct proliferative capacities, molecular signatures, developmental trajectories, and effector functions[70]. In contrast, killing-capable G-CSF-differentiated 32Dcl3 and HL-60 cells exist as terminally differentiated neutrophils and closely mimic the molecular and functional properties of mature neutrophils with potent bactericidal properties[37,71,72]. Sagiv et al. reported that while mature neutrophils are capable of exerting tumor cell killing, immature neutrophils are not[36]. In addition, not all mature neutrophils are killing-capable. Only high-density mature neutrophils can kill tumor cells, whereas low-density or immature (morphologically similar to low-density neutrophils) cannot[36]. Based on these previous studies, it is plausible that neutrophils isolated from the bone marrow under normal physiology (i.e., from a non-tumor-bearing animal) lack killing capacity, as the bone marrow houses neutrophils in a wide spectrum of maturation stages. During the

neutrophil-tumor cell interaction, we found that neutrophil-specific granules are transferred into tumor cells. These granules contain myeloperoxidase, which is required for neutrophils to induce ferroptosis. This observation is consistent with the early finding that neutrophils isolated from MPO-deficient patients lose cytotoxicity[16]. Although further studies are needed to delineate how MPO-containing granules are transferred into tumor cells, our observation of this phenomenon suggests that neutrophils may increase lipid peroxidation in tumor cells from certain intracellular compartments of tumor cells.

Necrosis usually develops along with tumor progression. The degree of tumor necrosis negatively correlates with GBM patient survival[8]. It is still unknown whether necrosis is an epiphenomenon of tumor progression or the cause of its aggressiveness. By defining GBM tumor necrosis as ferroptotic cell death, we were able to experimentally test these notions. Inhibiting ferroptosis by manipulating essential regulatory genes, such as *GPX4* and *ACSL4*, curtailed GBM aggressiveness and prolonged the survival of tumor-bearing mice (Fig. 6b, e). Of note, a GPX4 isoform (rGPX4) containing a mitochondrial targeting sequence was used in this study. It has been suggested that the cytosolic GPX4 isoform plays an essential role in ferroptosis prevention[73]. We observed that rGPX4 appeared to be in the cytosol and not limited to the mitochondria (Supplementary Fig. 6b). The nonspecific localization is probably due to either rGPX4 overwhelming mitochondrial protein transporters or to masking of the mitochondrial targeting sequence by the FLAG-HA tag on the rGPX4 amino-terminus (see Methods). The FLAG-HA tag did not appear to interrupt rGPX4 function in this case. This is consistent with the observation that GPX4 functionality is not affected by tags on its N-terminus[74]. Nevertheless, overexpression of rGPX4 might be a limitation of this study. Although it is still unclear how necrosis can decrease overall survival in both preclinical and clinical GBMs, we suspect that secretion of protumorigenic factors and/or recruitment and activation of other immune cells (e.g., microglia) by ferroptosed tumor cells may lead to certain non-autonomous tumor effects that precipitate neuroinflammation (or, more precisely, "necroinflammation") and cerebral cytokine storm, eventually leading to irreversible cerebral edema, cachexia, multi-organ dysfunction, and death, as observed in our GBM mouse model. Therefore, when advanced-stage tumors manifest necrosis, targeting ferroptotic cell death might benefit GBM patients by curtailing tumor necrosis-triggered sequelae.

## Methods

**Histologic neutrophil quantification in GBM patients**. Through Natural Language Search in Cerner, a list of deceased patients ($n = 132$) diagnosed with glioblastoma multiforme, WHO grade 4 on tumor resections performed at Penn State Hershey Medical Center between the years of 2009 and 2015 was obtained. Prior to further analysis, patient deaths were confirmed by electronic medical records on-file and/or available public records. Of these patients, only those with "prominent" or "extensive" necrosis specified in the pathology report were included for this part of the study ($n = 45$). Pathology glass slides and paraffin-embedded blocks of GBM tumors on the confirmed deceased patients were collected for neutrophil quantification. To capture the intratumoral heterogeneity, multiple bright field images of the largest areas of interest (tumor-infiltrating, interface, and necrotic regions) on H&E stained tumor slides were acquired with an Olympus CX41 microscope PLCN 40x objective lens for each patient. Tumor-infiltrating region was defined as where viable tumor cells infiltrate normal brain parenchyma. Necrosis region was defined as having no viable cells. Interface region was defined as the area interposed between the tumor-infiltrating region and adjacent necrotic region. On each slide, the largest regions of interest were used for image acquisition. At high-power (40x), images were taken from every other field to circumferentially evaluate each region of interest. Neutrophil counts were performed on all of the acquired images by a trained researcher under direct supervision of a pathologist. For images with discrepant neutrophil counts between researcher and pathologist, counts obtained in a blinded fashion by the senior pathologist were used for analysis. Using the chromogenic immunodetection of anti-human CD66b (method further described below), neutrophil counts were verified on three

randomly selected patients from this study. For survival analysis (Fig. 8g), all patients from this cohort are included ($n = 45$). Length of survival was calculated from the date of pathological diagnosis to the date of death. Patients were separated into two groups by median absolute neutrophil count/high-power field in necrotic region obtained from their diagnostic H&E-stained sections (as in Fig. 8c, e). The neutrophil high group consists of patients with absolute neutrophil count/field above the median (including the median; $n = 23$), whereas the neutrophil low group consists of patients with absolute neutrophil count/field below the median ($n = 22$). Only pre-existing data were obtained via review of electronic medical records (EMR), and therefore no further data collection or subject recruitment was conducted for this study. Tissue specimens and data collected from this decreased cohort were used in Fig. 8c–e. The study procedures and data collection were approved and considered non-human subject research by the Penn State Institutional Review Board (IRB).

**Radiographic volumetric analysis of GBM necrosis via post-contrast T1 MRI.** Subjects were selected from a cohort of patients seen in the Penn State Hershey Neuro-Oncology clinic between December 2018 and March 2019 (including those diagnosed and referred from outside facilities and those diagnosed here), and only patients with histologically confirmed grade 4 malignant gliomas were included in this study ($n = 75$). Pre-surgical, post-contrast axial T1-weighted fat-saturated (T1 FS) MRI images with a slice thickness of 5 mm from 75 patients with histologically confirmed GBMs were retrospectively analyzed. MRI images were acquired via standard multicontrast sequences including post-contrast fat-saturated T1 TSE sequence using either 1.5 T or 3.0T magnet (Siemens Healthcare) after injection of 0.1 mmol/kg of gadolinium (Gadavist, Bayer Schering Pharma). Regions of whole-tumor and central necrosis were manually traced and measured using Philips PACS (IntelliSpace PACS Enterprise 4.4) by a trained researcher under the supervision of a blinded neuroradiologist. Central necrosis was defined as non-enhancing areas within enhancing tumor with irregular inner margins on post-contrast T1-weighted images. Pre-surgical peripheral blood neutrophil counts were obtained from retrospective review of electronic medical records. For survival analysis (Fig. 8f), all deceased patients are included ($n = 37$). Length of survival was calculated from the date of pathological diagnosis to the date of death. Patients were separated into two groups by median radiographically identified necrosis-to-tumor volumetric ratios. The $N/T$ ratio high group consists of patients with necrosis-to-tumor ($N/T$) volumetric ratios higher than median ($n = 18$); $N/T$ ratio low group consists of patient with $N/T$ ratios lower than or including the median ($n = 19$). Only pre-existing data were obtained via review of electronic medical records (EMR) and imaging studies (MRI), and therefore no further data collection or subject recruitment was conducted for this study. The study procedures and data collection were approved by the Institutional Review Board (IRB) of Penn State Hershey Medical Center. Per the Penn State IRB, human subject research presented in Fig. 8b and Supplementary Fig. 8b–d in this manuscript was exempt from informed consent requirements.

**Mice and orthotopic xenograft tumor models.** Six-to-eight-week-old female athymic nude mice (Nu(NCr)-Foxn1nu Strain Code: 490, Charles River) were used for the GBM xenograft mouse models. For tumorigenesis experiments, human GBM cells were first transduced with a retroviral vector expressing firefly luciferase. These cells were then transduced with retroviral or lentiviral vectors expressing the indicated shRNAs or complementary DNAs (cDNAs). For each mouse, $3 \times 10^5$ cells were injected into the right hemisphere at coordinates (+1, +2, −3). Brain tumor growth was monitored with bioluminescence. Briefly, mice were anesthetized with isoflurane. 1.875 mg of luciferin dissolved in 125 μl of phosphate-buffered saline (PBS; 15 mg/ml) was intraperitoneally injected into mice. Ten minutes later, the mice were placed in the in vivo imaging system (IVIS) imaging chamber (Xenogen, Alameda, CA) and imaged for 1 min with the camera set at the highest sensitivity. Photons emitted from the brain region were quantified using LivingImage software (Xenogen). Luciferase activity was measured as photons emitted per second. For tumor sample preparation and histology, whole-brain tissue from tumor-bearing animals was fixed with 4% neutral-buffered formalin, embedded in paraffin, and submitted to Penn State College of Medicine's Comparative Medicine Histology Core, cut into sections 5 μm thick, and stained with hematoxylin & eosin (H&E). Areas of tumor and central necrosis were manually traced and measured with ImageJ for quantification of necrosis-to-tumor ratio. All animals were housed in a room with a 12-h light/dark cycle with free access to a standard rodent diet and water at ambient temperature maintained between 18 and 23 degrees Celsius and humidity between 40–60%. All experiments described in this study were carried out with the approval of the Penn State University Institutional Animal Care and Use Committee and in accordance with its guidelines.

**Quantification of necrosis-to-tumor ratio.** Paraffin-embedded, H&E-stained sections were used for all quantification of GBM necrosis-to-tumor ratio in the xenograft mouse model described above. Necrosis (N) is defined as acellular regions (appearing pale pink) within tumors as identified by H&E stain, and a cellular tumor (CT) region is defined as a hypercellular region. Regions of interest were manually traced and measured using the freehand tool in ImageJ. Given that the necrotic region would not distribute uniformly within a tumor, the necrosis-to-tumor ratio

could vary significantly in different sections of the same tumor. To make the quantification more accurate, 3–4 step sections (each 50 μm apart) obtained from the same tumor were used for comparison of necrosis-to-tumor area ratios, and the average of these ratios was used to represent each tumor (obtained from each tumor-bearing animal).

**Cells.** Human GBM cell line: LN229 (CRL-2611), U87 MG (HTB-14), and LN18 (CRL-2610); murine myeloblastic cell line, 32D Clone 3 (CR-11346); and human promyelocytic leukemia cell line, HL-60 (CCL-240), were purchased from ATCC. Human glioblastoma cells were cultured in Dulbecco's modified Eagle's medium (DMEM; 10-013-CV, Corning) supplemented with 10% fetal bovine serum (FBS; Gibco, 10437028) and 1% Antibiotic–Antimycotic Solution (30-004-CI, Corning) at 37 °C with 5% $CO_2$. 32D Clone 3 cells were cultured in Roswell Park Memorial Institute (RPMI) 1640 (10-040-CV, Corning) supplemented with 10% FBS and 1% Antibiotic–Antimycotic Solution as above with 5 ng/ml mouse interleukin-3 (IL-3) culture supplement (CB40040, ThermoFisher). HL-60 cells were cultured with RPMI 1640 supplemented with 20% FBS and 1% Antibiotic–Antimycotic Solution as above. The protocol for differentiating 32D Clone 3 cells into neutrophils was adapted from a previously described protocol[37]. Briefly, $2 \times 10^5$ cells/ml were seeded in a 55 cm$^2$ culture dish containing full media as above supplemented with 100 ng/ml of recombinant human granulocyte colony-stimulating factor (rhG-CSF; 214-CS, R&D) and cultured for 14 days. The protocol for differentiating HL-60 cells into neutrophils was adapted from previously described protocols[71,75,76]. Briefly, cells were cultured in RPMI 1640 supplemented with 10% FBS, 1% Antibiotic–Antimycotic Solution, 1 μM all-*trans*-retinoic acid (ARTA; R2625, Sigma), and 30 ng rhG-CSF for 21 days. Both 32D Clone 3 and HL-60 cells were supplemented with 1–2 ml of additional fresh full growth media with the aforementioned differentiation factors added every 3 days by adjusting the cells to maintain a density between $2 \times 10^5$ and $1 \times 10^6$ cells/ml; cell viability and morphological changes were checked accordingly. Differentiation was evaluated via flow cytometry by measuring Ly6G expression for 32D Clone 3 and CD66b (see antibody section for details) for HL-60. None of these cell lines were listed in the database of misidentified cell lines maintained by ICLAC and NCBI Biosample. These cell lines were not authenticated in this study. All cell lines were confirmed as *Mycoplasma* negative before experiments. Unless otherwise indicated, cells were grown to 50% confluence. For compound treatment experiments, cells (both monocultures and cocultures) were seeded in neutrophil complete growth medium without rhG-CSF or ARTA.

**Neutrophil isolation (brain, bone marrow, and spleen).** Mice were euthanized with carbon dioxide and cervical dislocation in keeping with the institution's animal care committee-approved protocol. To isolate tumor-associated neutrophils, intracranial tumors were resected from mice and washed three times with sterile-filtered Dulbecco's phosphate-buffered saline (DPBS; 21-030-CVR, Corning). Tumor tissue (ranging between 100-200 mg) was minced completely. Minced tissue was then incubated in 15-ml conical tubes containing Hank's Balanced Salt Solution (HBSS; 21-022-CV, Corning) with 0.1 mg/ml type IV collagenase (C5138, Sigma) and 0.1 mg/ml hyaluronidase (H6254, Sigma) at 37 °C for 15 min. Digestion was stopped by adding equal volumes of 2% FBS/HBSS mixture. Cells were then centrifuged for 5 min at 1000 rpm (110 × $g$; ST-40 centrifuge, ThermoFisher) and washed twice with 2% FBS/HBSS mix to completely remove digestive enzymes and then filtered through a sterile 40 μm nylon mesh cell strainer (22-363-547, ThermoFisher). Neutrophil isolation from bone marrow and spleens of non-tumor-bearing mice was adapted from protocols previously described[77,78]. For bone marrow extraction, cells were isolated from one femur and one tibia of each mouse via centrifugation at 400 × $g$ for 7 min at 4 °C following incision of bone epiphyses. For spleen neutrophil isolation, the whole spleen of one mouse was used for each sample. Cells were dissociated by crushing the spleen using an insulin syringe plunger against a sterile 40 μm nylon mesh cell strainer over a Petri dish. Red blood cells were lysed from tumor, bone marrow, and spleen single-cell suspensions with red blood cell lysis buffer (11814389001, Sigma), washed three times with RPMI with 2% FBS, loaded with trypan blue, counted with an automated cell counter, and resuspended in DPBS with 2% FBS prior to proceeding to either immunomagnetic selection or flow cytometry. Viability of the isolated cells was evaluated via flow cytometry using cell nucleic acid fluorescent dye, Sytox-Green (ThermoFisher). Percent viability was plotted as Sytox-Green⁻ cells out of total single cells with LN229 parental cells obtained from regular culture as negative control.

**Immunomagnetic selection and immune profiling via flow cytometry.** Immunomagnetic selection of mouse neutrophils from the tumor, spleen, and bone marrow was performed using a mouse neutrophil isolation kit (130-097-658, Miltenyi Biotec) following the manufacturer's instructions. Briefly, following cell dissociation as described above, singly suspended live cells were first blocked with purified rat anti-mouse CD16/32 Fc receptor block (553141, BD Biosciences) prior to antibody labeling. For immunomagnetic selection, dissociated cells were labeled with biotin-coupled anti-mouse Ly6G, followed by positive magnetic selection using anti-biotin microbeads and MS columns (130-042-201, Miltenyi Biotec). The purity of the total neutrophil population was typically higher than 90% as evaluated via flow cytometry. Viability of the immunomagnetically isolated neutrophils was

evaluated by flow cytometry using cell nucleic acid fluorescent dye, Sytox-Green (ThermoFisher). Percent viability was plotted as Sytox-Green⁻ cells out of total single cells with LN229 parental cells obtained from regular culture as negative control. For flow cytometry, live single-cell suspensions at a concentration of $1 \times 10^6$ cells/ml prepared from fresh tumor tissue as described above were first blocked with anti-mouse CD16/32 Fc receptor block followed by surface labeling of anti-CD45 (clone 30-F11), anti-CD11b (clone M1/70), and anti-Ly6G (clone 1A8) antibodies at room temperature for 20 min. Details of antibodies used are outlined in the antibody section below. Cells were then washed three times, resuspended in 1 ml of DPBS, and run on an LSRFortessa (BD Biosciences) cell analyzer in the Penn State College of Medicine's Flow Cytometry Core.

**Ly6G-mediated neutrophil depletion.** Xenograft GBM tumor implantation of LN229 stably expressing TAZ⁴ˢᴬ was performed as detailed above. Bioluminescence imaging (BLI) was performed 2 weeks post implantation to verify location of tumor growth. Mice were then randomly assigned into two treatment groups based on their BLI readings, with one group receiving monoclonal Ly6G antibody and the other receiving isotype controls. To start depleting neutrophils prior to necrosis formation, mice were injected intraperitoneally with 50 µg of *InVivo*Plus rat anti-mouse Ly6G, clone 1A8 (BE0075-1, BioXCell) or *InVivo*Plus rat IgG2a isotype control, clone 2A3 (BE0089, BioXCell) in a volume of sterile 100 µl DPBS every 3 days, from 2 weeks post implantation onward. Depleting antibodies or isotype controls were administered every 2 days starting at 3 weeks post implantation and daily starting at 4 weeks post implantation. Effectiveness of neutrophil depletion was monitored via peripheral blood smears and flow cytometry of whole-tumor tissue.

**Cell viability assays via luminescence.** Luciferase-expressing human GBM cells (LN229) stably expressing TAZ⁴ˢᴬ were seeded in 96-well flat-bottom plates at a density of 3000 cells/well alone or cocultured with neutrophils in a 1-to-20 ratio in duplicates and incubated at 37 °C for 15 h. At the end of coculture, luminescence was measured as a readout for cell viability using the Luciferase Assay System (E1500, Promega) following the manufacturer's instructions. Briefly, cells were washed once with DPBS, lysed, and given a D-firefly luciferin potassium salt dissolved in culture medium prior to luminescence measurement via a multi-mode plate reader (BMG Labtech). Luminescence from tumor cell monoculture was used to establish a baseline reading. All readings were averaged between duplicate wells. Percent survival was calculated by normalizing blank-subtracted luminescence of coculture wells to that of tumor cell monoculture. For all compound treatment experiments, tumor cell monoculture treated with the same compound at the same concentration was used as control for normalization (100% survival).

**Colony-formation assay.** Colony-formation assays to assess tumor clonogenic survival following neutrophilic coculture were conducted as follows[27]. Briefly, LN229 cells stably expressing TAZ⁴ˢᴬ and GFP were washed once with DPBS, trypsinized, and subject to fluorescence-activated cell sorting (FACS) in a FACSAria SORP high-performance cell sorter (BD Biosciences) in the Penn State College of Medicine Flow Cytometry Core after 48-h coculture with mouse (32D Clone 3) or human (HL-60) neutrophils at a one-to-five ratio in full neutrophil growth media as above. After sorting, 400 GFP⁺ tumor cells were seeded in complete medium in each well of a 6-well plate and cultured for 10–12 days. Plates were then washed with PBS and stained with crystal violet (6101, ENG Scientific Inc) to visualize colonies. Colony images were scanned and quantified in ImageJ. Next, to assess viability among tumor cells containing intracellular neutrophilic granular contents, mouse neutrophils (32D Clone 3) were labeled with lipophilic PKH26 red fluorescent cell linker dye (PKH26GL-1KT, Sigma) before coculture with GFP⁺ TAZ⁴ˢᴬ-expressing tumor cells as described above. Following coculture and cell sorting as above, 400 tumor cells doubly positive for green and red fluorescence were seeded with green-only tumor cells from the same coculture condition as control, and they then underwent the colony-formation culture procedure described above.

**Cell viability assays via nucleic acid fluorescent dye.** mCherry-expressing LN229^TAZ(4SA) tumor cells were seeded in 12-well flat-bottom plates at a density of 50,000 tumor cells/well alone or cocultured with unlabeled neutrophils in a 1-to-5 ratio in duplicates and incubated at 37 °C. Following 2-day coculture, cells were loaded with live cell-impermeant nucleic acid stain, Sytox-Green (S34680, ThermoFisher) or Sytox-Blue (S34857, ThermoFisher), as a dead cell indicator to determine cell viability via flow cytometry following the manufacturer's instructions. Briefly, at the end of coculture, cells were harvested, washed three times with cold DPBS at a concentration of $1 \times 10^6$ cells/ml, and labeled with Sytox-Green or Blue at a 1-to-1000 dilution. Labeled cell mixtures were then incubated at room temperature for 20 min and protected from light prior to flow cytometry analyses using a blue 488 nm laser (for Sytox-Green) or violet 405 nm laser (for Sytox-Blue). Survival is plotted as percent of mCherry^high cells in each condition gated by Sytox-Blue signals in LN229^TAZ(4SA) tumor cells cultured alone (as 100%). For cell viability comparison of PKH^high⁻ and PKH^low/⁻-LN229^TAZ(4SA) cells, ZsGreen-expressing LN229^TAZ(4SA) cells were seeded alone or cocultured with neutrophils prelabeled with lipophilic PKH26 red fluorescent cell linker dye as described above.

Following coculture, cells were loaded with live cell-impermeant nucleic acid stain, Sytox-Blue, as above, to examine cell viability. All flow cytometry analyses were run using an LSR Fortessa (BD Biosciences) cell analyzer in the Penn State College of Medicine's Flow Cytometry Core and analyzed using FlowJo software V10 (BD Biosciences).

**Immunoprecipitation and immunoblotting.** For FLAG-mediated immunoprecipitation, $3 \times 10^6$ indicated cells containing FLAG-tagged proteins were seeded in a 10 cm plate overnight and lysed in RIPA buffer (50 mM Tris, pH 7.5, 150 mM NaCl, 4 mM EDTA, 1 mM EGTA, 1% Triton X-100, 0.5% sodium deoxycholate, 0.1% sodium dodecyl sulfate, 10% glycerol, 1x phosphatase inhibitor (Roche) and 1x proteinase inhibitor (Roche)). Cell lysates were incubated with anti-FLAG M2 Affinity Gel (Sigma) at 4 °C for 2 h. The precipitates were washed three times with RIPA buffer and eluted with 3xFLAG Peptide (Sigma # F4799) dissolved in TBS (10 mM Tris HCl, 150 mM NaCl, pH 7.4). Immunoblotting was conducted as follows[27]. Briefly, cells were lysed in SDS lysis buffer (10 mM Tris pH 7.5, 1% SDS, 50 mM NaF, 1 mM Na₃VO₄) and subjected to sodium dodecyl sulfate–polyacrylamide gel electrophoresis (SDS–PAGE) on 4–12% Bis-Tris SDS–PAGE gels (Invitrogen) and transferred to Immobilon-P membranes (Millipore). Membranes were incubated in blocking buffer (5% skim milk/TBST [0.1% Tween, 10 mM Tris at pH 7.6, 100 mM NaCl]) for 1 h at room temperature and then with primary antibodies diluted in 5% bovine serum albumin (BSA)/TBST overnight at 4 °C. After three washes, membranes were incubated with goat anti-rabbit HRP-conjugated antibody or goat anti-mouse HRP-conjugated antibody (7074S and 7076S, Cell Signaling Technologies; 1:5000 for both) at room temperature for 2 h and subjected to chemiluminescence using ECL (1856136, Pierce).

**Immunofluorescent staining and chromogenic immunodetection.** Immunofluorescent staining was performed as follows[27]. For histological samples, paraffin-embedded 5-µm sections were deparaffinized and rehydrated in successive baths of xylene and ethanol (100%, 95%, 70%, and 50%), followed by heat-induced (95 °C) epitope retrieval in 10 mM sodium citrate buffer (pH = 6.0). After 1 h block with 5% BSA/PBS at room temperature, samples were incubated overnight at 4 °C with primary antibodies diluted in 2.5% BSA/0.05% Triton X-100/PBS. The next day, sections were washed three times with 0.1% Triton X-100/PBS prior to incubation with secondary antibody diluted in 2.5% BSA/0.05% Triton X-100/PBS for 60–90 min at room temperature. Then, sections were again washed three times with 0.1% Triton X-100/PBS, labeled with 4,6-diamidino-2-phenylindole (DAPI) for nuclear visualization, rinsed with PBS, and mounted in ProLong Gold Antifade Mountant (P10144, Invitrogen). Chromogenic immunodetection on histological samples was conducted using an Anti-Rabbit HRP-DAB Cell & Tissue staining kit following the manufacturer's instructions (CTS005, R&D) after deparaffinization and blocking as above. All primary antibodies were diluted at a 1-to-100 concentration and all secondaries at 1-to-200 concentration unless otherwise specified.

**PKH26/MPO puncta quantification and immunocytochemistry (ICC).** ZsGreen-expressing LN229^TAZ(4SA) tumor cells (16,000/well) were seeded on glass coverslips one day prior to coculture either alone or with various types of neutrophils labeled with the lipophilic PKH26 red fluorescent cell linker dye described above in a 1-to-3 ratio for 48 h at 37 °C. Following coculture, cells were washed once with DPBS, fixed with 4% paraformaldehyde in PBS for 20 min at room temperature, and incubated in permeabilization buffer (PDT: 0.3% sodium deoxycholate, 0.3% Triton X-100 in PBS) for 30 min on ice. Afterwards, cells were stained with anti-mouse myeloperoxidase (MPO) antibody (see the antibody section for details) overnight at 4 °C, incubated with secondary antibody, labeled with DAPI, and mounted as described above. All primary antibodies were diluted at a 1-to-100 concentration and all secondaries at 1-to-200 concentration unless otherwise specified. Stained cells were then imaged at 0.5 µm increments along the z-axis of a Leica SP8 inverted confocal laser scanning microscope at Penn State College of Medicine's Light Microscopy Core. Optical excitation was carried out with a 405 nm diode laser and 488, 561, and 633 nm argon lasers. Images were then stacked with maximal intensity and merged using ImageJ 1.50, and fluorescent puncta were quantified using the Analyze Particles function in ImageJ. Tumor cells were outlined using the GFP channel, and only puncta (for both PKH26 and MPO) located within tumor cells were included for quantification. To test whether MPO puncta formation within tumor cells requires direct cell–cell contact, LN229^TAZ(4SA) tumor cells were seeded in 24-well flat-bottom plates containing glass coverslips as described above. Murine (32Dcl3) or human (HL-60) neutrophil-derived conditioned media (CM) were collected from two-day 32Dcl3 or HL-60 monoculture containing 9 times the number of tumor cells seeded on coverslips. Upon CM collection, supernatants obtained from either 32Dcl3 or HL-60 monoculture were first centrifuged at $110 \times g$ (ST-40, ThermoFisher) for 4 min to eliminate cellular debris, followed by 1:2 dilution in fresh culture media prior to addition to LN229^TAZ(4SA) tumor cells pre-seeded on coverslips. In our second approach, ZsGreen-expressing LN229^TAZ(4SA) tumor cells were seeded on coverslips one day prior to coculture as above. The next day, differentiated murine (32Dcl3) or human (HL-60) neutrophils were seeded as above with or without the cell-impermeable 0.4 µm pore size polyethylene terephthalate Millicell hanging cell culture insert (EMD Millipore) into the 24-well plates containing pre-seeded tumor cells.

Following two-day coculture, cells were stained with anti-human/mouse MPO as in the ICC procedure above. Stained cells were then imaged using the Olympus CX41 microscope PLCN 40x objective and quantified as above via ImageJ.

**Quantification of endothelial cells, astrocytes, and microglia on immunofluorescent IHC**. Immunofluorescent staining on paraffin-embedded 5-μm brain tumor sections was performed as described above. For endothelial cell analysis, CD31 was used as an endothelial marker. Image acquisition was performed within 1–3 days following IHC using an Olympus CX41 microscope PLCN 20x objective to cover the whole section. Images were stitched using the automated photomerge function on the microscope software. Stitched whole-section images were then converted to grayscale 8-bit images, thresholded, and quantified using the Analyze Particles function in ImageJ. Data were plotted as total CD31$^+$ cell area normalized to each field area (i.e., intratumor vs. non-tumor parenchyma). DAPI images were used to distinguish intratumor vs. non-tumor parenchyma. For astrocyte analysis, GFAP was used as a marker for astrocytes. Due to intratumoral heterogeneity of GFAP signals and to avoid selection bias, intratumoral images were taken from regions with the highest GFAP signal intensities (at the edge of cellular tumor) in every other high-power field (40x objective) to circumferentially evaluate regions of interest. GFAP signals in non-tumor parenchyma appeared largely homogeneous, and all images of non-tumor parenchyma were taken from the cortices of the non-tumor-containing hemispheres. For microglia analysis, TMEM119 was used as a microglia marker. Owing to intratumoral heterogeneity of TMEM119 signals and to avoid selection bias, intratumoral images were taken from regions with the highest TMEM119 signal intensities (along the necrotic regions within each tumor) in every other high-power field (40x objective) to circumferentially evaluate regions of interest. TMEM119 signals in non-tumor parenchyma appeared largely homogeneous, and all images of non-tumor parenchyma were taken from the cortices of the non-tumor-containing hemispheres. All images were first converted to grayscale 8-bit images prior to thresholding and particle analysis using ImageJ. Corresponding DAPI images were again used to distinguish intratumor vs. non-tumor parenchyma and for quantification of field area in each image. Data were plotted as numbers of total GFAP$^+$ or TMEM119$^+$ cell objects normalized to each image field area.

**In situ Liperfluo dye loading**. To detect accumulation of lipid peroxides in live tumor tissue, in situ Liperfluo dye loading was conducted following the manufacturer's instructions (L248-10, Dojindo Molecular Technologies, Inc). Briefly, immediately after euthanasia as described above, the cerebrum was excised, quickly mounted on a vibratome (Vibratome 1000, IMEB Inc.), and submerged rapidly in 4 °C oxygenated Krebs solution for sectioning. The tumor-laden brain tissue was sliced into 300-μm sections and placed in containers filled with 30 °C oxygenated Krebs solution until Liperfluo dye was loaded. Sections submerged in Krebs solution were loaded with Liperfluo (suspended in DMSO, final concentration 10 μM) using a free floating immunolabeling protocol in 24 multi-well plates at 37 °C for 4 h with occasional gentle shaking. Following dye labeling, sections were washed in DPBS three times with gentle shaking in the dark at room temperature and placed on a glass slide for microscopy. Then, brain slices were fixed with 2% paraformaldehyde and labeled with DAPI for nuclear visualization.

**Intracellular lipid peroxides and ROS detection via BODIPY-C11 and CM-H2DCFDA**. Intracellular lipid peroxides were detected via BODIPY 581/591 C11 (D3861, ThermoFisher Scientific), and intracellular ROS were detected via CM-H2DCFDA (C6827, molecular probes, Invitrogen) following manufacturers' instructions. To identify the type of cells (i.e., tumor vs. immune cells) with intracellular accumulation of lipid peroxides and ROS, single-cell suspensions prepared from fresh brain tumor tissues as above were first labeled with anti-CD45 (clone 30-F11) as described above followed by incubation in 1 ml HBSS containing 5 μM BODIPY 581/591 C11 or 2 μM CM-H2DCFDA for 15 min at 37 °C in a tissue culture incubator with occasional shaking. Following surface labeling and dye loading, cells were washed three times with DPBS, resuspended in 200 μl fresh HBSS, and then analyzed immediately with the BD LSRFortessa (BD Biosciences) instrument at the Penn State College of Medicine's Flow Cytometry Core Facility.

**Transmission electron microscopy**. Brain tumor tissues from mice in the above xenograft model were dissected, rinsed once with cold PBS, and immediately fixed in half-strength Karnovsky fixative containing 2% paraformaldehyde and 2.5% glutaraldehyde (pH 7.3) at 4 °C overnight. For in vitro coculture samples, $8 \times 10^5$ LN229 tumor cells stably expressing TAZ$^{4SA}$ were seeded in a sterile 60 ×15 mm Permanox dish (Nalge Nunc International) one day prior to coculture with either in vitro differentiated mouse neutrophils (32D Clone 3) or human neutrophils (HL-60) in a 1-to-10 ratio for 12 h at 37 °C. Tumor cells in monoculture served as controls. Cells were washed twice with warm PBS and fixed with the Karnovsky fixative as described above at room temperature for an hour. Following fixation, samples were submitted to the Transmission Electron Microscopy Core for processing. Briefly, samples were further fixed in 1% osmium tetroxide in 0.1 M sodium cacodylate buffer (pH 7.4) for 1 h. Samples were dehydrated in a graduated ethanol series followed by acetone and embedded in LX-112 (Ladd Research, Williston, VT). Thin sections (60 nm) were stained with uranyl acetate and lead citrate and viewed in a JEOL JEM1400 Transmission Electron Microscope (JEOL USA Inc., Peabody, MA).

**Human cytokine array**. An unbiased human cytokine screening was carried out using the ELISA-based 80-target cytokine array kit following the manufacturer's instructions (ab133998, Abcam). Briefly, whole-tumor tissues were dissected out from LN229$^{vector}$ and LN229$^{TAZ(4SA)}$ tumor-bearing mice upon reaching endpoints and homogenized in lysis buffer (provided in the kit) via a tissue homogenizer. Total lysate protein concentrations from all samples were first quantified using the BCA method and diluted according to the manufacturer's recommendation. For each sample, 200 μg of total protein in 1 ml of blocking buffer (provided in the kit) was used for each array membrane.

**Antibodies, reagents, compounds**. *Antibodies for flow cytometry:* FITC anti-mouse CD11b (101205, Biolegend, dilution 1/100), PE anti-mouse CD45 (103105, Biolegend, dilution 1/100), BV421 Rat anti-mouse CD45 (563890, BD Biosciences, dilution 1/100), and anti-mouse CD16/32 Fc receptor block (cat#553141, lot#9060742, BD Biosciences, 0.25 μg/10$^6$ cells).

*Antibodies for IHC, ICC, and chromogenic immunodetection:* Ultra-LEAF purified rat anti-mouse Ly-6G (1A8, 127620, Biolegend), rabbit anti-mouse/human myeloperoxidase (ab9535, Abcam), anti-mouse/human PTGS2 (clone H-3, sc-376861, Santa Cruz), anti-mouse/human Cox2 (D5H5) XP (cat#12282 S, lot#5, Cell Signaling Technologies), mouse anti-human CEACAM8/CD66b (G10F5, Novus Biologicals), rabbit anti-glutathione peroxidase 4, GPX4 (ab125066, Abcam), N-cadherin (13-A9, sc-59987, Santa Cruz Biotechnology), anti-mouse CD31 (89C2, 3528S, Cell Signaling Technologies), rabbit anti-human/mouse GFAP (D1F4Q, 12389S, Cell Signaling Technologies), rabbit anti-mouse TMEM119 (28-3, ab209064, Abcam), and mouse-anti-HA.11 (16B12, MMS-101R, Biolegend). For IHC and ICC, all primary antibodies were used at a 1/100 dilution and secondary antibodies at a 1/200 dilution).

*Antibodies for immunoblotting:* Mouse anti-human fibronectin (clone EP5; Santa Cruz, sc-8422, dilution at 1/5000), rabbit anti-human CD44 (ab157107, Abcam, dilution at 1/10,000), rabbit anti-human CTGF (86641, Cell Signaling Technologies, dilution at 1/5000), mouse anti-β-Actin (3700, Cell Signaling Technologies, dilution at 1/20,000), GPX4 (clone E-12, sc-166570, Santa Cruz, dilution at 1/1000), TAZ (V386, 4883S, Cell Signaling Technologies, dilution at 1/1000), anti-human ACSL4 (F-4, sc-365230, Santa Cruz Biotechnology, dilution at 1/1000), anti-mouse/human MPO (ab9535, Abcam, dilution at 1/1000), anti-human MPO (E1E7I, 14569, Cell Signaling Technologies, dilution at 1/1000) and anti-human GAPDH (0411, sc-47724, Santa Cruz Biotechnology, dilution at 1/5000), goat anti-rabbit HRP-conjugated antibody (7074S, Cell Signaling Technologies, dilution at 1/5000), and goat anti-mouse HRP-conjugated antibody (7076S, Cell Signaling Technologies, dilution at 1/5000).

*Compounds and final concentrations for inhibitor screening:* z-VAD-fmk (HY-16658, MedChem Express, 25 μM), Necrostatin-1 (11658, Cayman Chemicals, 30 μM), GSK'872 (530389, EMD, 15 μM), Necrosulfonamide (20844, Cayman Chemicals, 1 μM), Ferrostatin-1 (SML0583, Sigma, 2 μM), Liproxstatin-1 (SML1414, Sigma, 0.2 μM), N-acetylcysteine (A7250, Sigma, 5 mM), Deferoxamine mesylate salt (D9533, Sigma, 0.1–0.2 mM), PF06282999 (PZ-0375 5MG, Sigma 2 μM), and 4-Aminobenzoic hydrazide, 4-ABAH (103200050, Acros Organics, 2 μM). All compounds above, except N-acetylcysteine (only soluble in water), were dissolved in dimethyl 2-oxoglutarate (DMSO; 349631, Sigma) prior to treating cells.

**Gene expression and silencing**. pBabe-neo-*TAZ* (4SA)[26] was generously provided by Dr. Kun-Liang Guan. pBabe-neo-*TAZ* (4SA-S51A) was generated by site-directed mutagenesis using the Quickchange mutagenesis kit (Stratagene). MGC Human *GPX4* Sequence-Verified cDNA (MHS6278-202801819) was purchased from Dharmacon and subcloned into the pBabe-puro vector with a FLAG-HA-tag on the amino-terminus to generate pBabe-puro-FLAG-HA-*GPX4*. Lentiviral vectors encoding shRNAs targeting *ACSL4* (#41: TRCN0000045541; #42: TRCN0000045542) were used to generate *ACSL4* knockdown in LN229$^{TAZ(4SA)}$. Lentiviral vectors encoding shRNAs targeting mouse *MPO* (Sh-*MPO*#45: TRCN0000076745; Sh-*MPO*#46: TRCN0000076746) and human *MPO* (Sh-*MPO*#03: TRCN0000046003; (Sh-*MPO*#04: TRCN0000046004) were used to generate *MPO* knockdown in differentiated murine (32Dcl3) or human (HL-60) neutrophil-like cells. All shRNA-expressing positive clones were selected via 2 μg/ml puromycin diluted in culture media. Positive clones were then cultured and expanded in puromycin-containing (2 μg/ml) differentiation media (as described above). Knockdown efficiency on positive clones was verified via western blotting once per week following selection. All shRNAs were obtained from the Penn State shRNA library core facility.

**Gene expression dataset analysis**. A GBM gene expression dataset was downloaded from the Ivy Glioblastoma Atlas Project (https://glioblastoma.alleninstitute.org/). For genes differentially expressed between the tumor peri-necrotic zone (PNZ) and the cellular tumor region (CT), the comparison was performed through the website directly. For Ingenuity Pathway Analysis, 1.5-fold and $P < 0.05$ were used as the cutoffs for differentially expressed genes. Indirect relationships were

chosen. Gene Set Enrichment Analysis (GSEA) was performed using GSEA 4.0 software provided by the Broad Institute (http://software.broadinstitute.org/gsea/index.jsp). The gene sets used for the analysis were generated by surveying published studies showing genes involved in ferroptosis[19,20]. These genes were classified into ferroptosis-promoting (positive role) or -inhibiting (negative role) genes based on the reference therein.

**Statistics and reproducibility**. For statistical analyses (including animal studies), samples sizes were chosen based on whether differences between groups are biologically meaningful and statistically significant. No data were excluded from the analyses. For cell experiments, all cells in each experiment were from the same pool of parental cells. All mice in each experiment were from the same cohort. The mice were randomly chosen for implantation of different types of cells. For data collected by objective instruments, such as plate readers, qPCR cyclers, microscopy software, flow cytometers, animal IVIS systems, and western blotting, the investigators were not blinded to group allocation during data collection. However, investigator bias is not considered to contribute to the data. For animal studies, the investigators were not blinded to group allocation during data collection when using the above mentioned objective instruments, but they were blinded during data analyses. Additionally, for all animal studies, randomization occurred in a blinded fashion. For clinical data analyses, investigators were not blinded during group allocation but were blinded during data analyses. Collected data were then verified by trained physicians who were blinded to group allocation. Statistical significance was determined as indicated in the figure legends. All center values shown are mean values, and all error bars represent standard errors of the mean (s.e.m). All statistical calculations and plotting were performed using GraphPad Prism 8. For all in vivo experiments, each data point represents an animal. For all in vitro experiments, each data point represents an average of technical replicates obtained from an independent experiment. The reproducibility of experiments (denoted by N) in each main figure is detailed in the corresponding legend and is summarized as number of independent experiments out of number of similar results as follows. For Fig. 1d, e, $N = 6/6$ for LN229$^{TAZ(vector)}$, $N = 5/5$ for LN229$^{TAZ(4SA)}$, and $N = 3/3$ for LN229$^{TAZ(4SA-S51A)}$. For Fig. 2a, b, and g, $N = 3/3$. For Fig. 4e, $N = 2/2$. For Fig. 4f, g, i, and j, $N = 3/3$. For Fig. 5f,g, $N = 3/3$. For Fig. 6a, $N = 3/3$. For Fig. 8c, $N = 45/45$. For Fig. 8e, $N = 3/3$. The reproducibility of each experiment shown in supplementary figures (denoted by N) is summarized as follows. For Supplementary Fig. 1c, f, and g, $N = 3/3$. For Supplementary Fig. 2d, $N = 3/3$. For Supplementary Fig. 3g and m, $N = 3/3$. For Supplementary Fig. 3h, $N = 3/3$, except for B.M. neutrophils, $N = 2/2$. For Supplementary Fig. 3j,k, $N = 4/4$. For Supplementary Fig. 4a, $N = 4/4$ for alone, and $N = 3/3$ for each of the other conditions. For Supplementary Fig. 4c, $N = 2/2$ for HL-60, and $N = 3/3$ for each of the other conditions. For Supplementary Fig. 4d and g, $N = 3/3$. For Supplementary Fig. 4f, $N = 4/4$ for TANs, and $N = 5/5$ for each of the other conditions. For Supplementary Fig. 5a–c, f, and g, each data point represents an average of an independently examined image field. For Supplementary Fig. 5a–c, an image field contains an average of 10–15 cells; images were taken from three independent experiments or animals; $N = 3/3$. For Supplementary Fig. 5f, g, an image field contains an average of 80–100 cells; images were taken from two independent experiments; $N = 2/2$. For Supplementary Fig. 5e, h, and i, $N = 3/3$. For Supplementary Fig. 6f, $N = 2/2$. For Supplementary Fig. 6h, $N = 3/3$ for each condition. Experiments in Supplementary Fig. 6a, b, and g were performed once. Results in all experiments that were not repeated were highly consistent with the results from other experiments in this study: Supplementary Fig. 6a, b directly support each other, and Supplementary Fig. 6f, g directly support each other. Further details of statistics and study reproducibility can be found in the source data sheet.

**Reporting summary**. Further information on research design is available in the Nature Research Reporting Summary linked to this article.

## Data availability

The publicly available datasets used in this manuscript were downloaded from the Ivy Glioblastoma Atlas Project (https://glioblastoma.alleninstitute.org/) or cBioPortal (https://www.cbioportal.org/). Source data are provided with this paper. All remaining relevant data are available in the article, Supplementary information, or from the corresponding author upon reasonable request.

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

## Acknowledgements

We would like to thank Dr. Kun-liang Guan for reagents; members of the Li Laboratory for helpful discussions; Dr. Kristin Browning from the Department of Neural and Behavioral Sciences for equipment sharing; Ms. Kristin Shuler and Mr. John Graybeal from the Department of Neurosurgery's Neuroscience Research Institute for assistance with IRB submissions; Dr. Han Chen from the Transmission Electron Microscopy Core; Mr. Nate Sheaffer, Ms. Jade Vogel, and Mr. Joseph Bednarczyk from the Flow Cytometry Core; Dr. Thomas Abraham and Mr. Wade Edris from the Microscopy Imaging Core (Leica SP8 Confocal: 1S10OD010756-01A1 CB); Dr. Sang Lee from the Bioluminescent Imaging Core; Dr. Teodora Orendovici and Dr. Yuka Imamura from the Genomics Sciences Core; Dr. Katherine Aird from the shRNA library core; Ms. Gretchen Snavely and Ms. Erin Mattern from the Comparative Medicine Histopathology Core; Ms. Jessica Wingate from the Comparative Medicine Diagnostic Laboratory; Ms. Marianne Klinger from the Department of Pathology's Molecular and Histopathology Core at Penn State College of Medicine for technical support and biospecimen processing; and Dr. Christopher Yee for editorial assistance. We acknowledge support from the National Institutes of Neurological Disorders and Stroke (R01 NS109147 to W.L.), Meghan Rose Bradley Foundation (214389 to W.L.), American Cancer Society Institutional Research Grant (124171-IRG-13-043-01 to W.L.), the Four Diamonds Fund for Pediatric Cancer Research (to PSU), and Penn State College of Medicine Medical Scientist Training Program (5T32GM118294 to P.Y. through PSU).

## Author contributions

P.Y. and W.L. conceived the project and designed all experiments. P.Y. performed all flow cytometry, cell sorting, electron microscopy studies, statistical analyses, and preparation of figures with assistance from W.L. P.Y. executed most in vitro experiments with assistance from Y.W., T.L., M.Y., and W.L. P.Y. and W.L. performed all stereotaxic intracranial surgeries in mice. Y.W., S.K., S.C., M.T., Z.L., and B.A. assisted mice-related experiments. P.Y. performed all human histopathological and clinical sample analyses under the supervision of C.L. and C.S. with verification of counts done by C.L. P.Y. and S.K. performed immunohistochemistry on human and mouse brain tumor specimens. P.Y. performed human radiographical analyses and electronic medical chart searches under the supervision of K.T. W.L. performed gene expression dataset analysis. B.Z., D.A., M.G., and C.S. provided lists of GBM patients seen at the Neurooncology clinic at Penn State Hershey Medical Center. C.S. provided clinical biospecimens, and H.W. provided biomaterials. P.Y. and W.L. wrote an original manuscript. All authors provided intellectual input and edited the manuscript. W.L. supervised all aspects of the work.

## Competing interests

The authors declare no competing interests.
