## [Peer Review File · Nature Communications]

REVIEWER COMMENTS

Reviewer #1 (Remarks to the Author): with expertise in glioblastoma

Major aspects

1. The selection of cell lines used should be explained. As the molecular background (p53 status etc.) may be critical for the phenomenon observed, these details need to be provided and evaluated.
2. The different genetic constructs for TAZ are well chosen.
3. The change between labeling the infiltrating cells neutrophils and broader CD11b+CD45+ myeloid cells should be made clearer. Also in the analysis of the temporal correlation of tumor recruitment of inflammatory cells.
4. As it has been reported that tumors are attractive to myeloid/progenitor/stem cells, it should be closer carved out, which cells are present at which stage in which quantities.
5. The time courses for the depletion experiments in Figure 3 are not fully transparent. Early/late is well understood, but the respective times are not matching.
6. The conclusion that a reduced necrotic area in Lys6-treated mice is due to reduced neutrophil numbers has an alternative, that is a direct impact of Lys6 on the necrosis development (i) by local effects on the resident immune cells or (ii) alternative tumor cell-directed mechanism.
7. Data on the cytotoxicity of neutrophils from different species are mainly correlative, but not sufficiently proven.
8. The reference for the survival rates as reported in Fig 4a,b are not fully clear. It seems that this presentation has been chosen to avoid full transparency on the actual numeric values. Are the data from one experiment and if so, absolute values and respective error bars would be a preferred mode of presentation.
9. Similar, as the experiments (in vitro and ex vivo) in 4 a and 4b are different, appropriate ways to depict the different data are valuable.
10. Data on MPO transfer and inhibition in Figures 5/6 are convincing.
11. The clinical data are meant to transfer the experimental findings to the patient site. Whereas the correlation between neutrophil signatures and tumor subtypes are interesting (see below), the attempt to correlate pretumor diagnosis circulating neutrophil numbers with tumor/necrosis ratio is prone to false conclusions due to the mobility of the compartment, impact of infections, medication (steroid plus/minus) etc. The used concept of tumor subtypes, which is commonly used relies on the stability and ignores the multiple subtypes in one summary subtype as published in a very elegant study from the MGH.
12. The proposed clinical applicability of the findings would need some more consideration of the appropriate timing of an intervention in the real world setting.

Minor aspects

1. TAZ needs to be explained in the abstract.

Reviewer #2 (Remarks to the Author): with expertise in ferroptosis

Tumor necrosis is commonly observed in poorly vascularized solid tumors, although the mechanisms how this is brought about remain poorly understood. In this work, the authors studied whether tumor infiltrating neutrophils contribute to the necrotic core by triggering ferroptosis in glioblastoma cells, both in cultured cells and intracranially implanted tumors. The authors provide evidence that the transfer of myeloperoxidase (MPO) from neutrophils to tumor cells may be one of the underlying mechanisms how ferroptotic cell death determines the extent of tumor necrosis, survival of tumor bearing mice and GBM patient prognosis. While the study is of interest, there are a number of uncertainties and open questions that call for a careful revision.

Comments:

- Fig. 1: The authors should include immunoblot data on exogenously expressed TAZ
- Fig. 3/4: Are some of the effects (e.g. neutrophil mediated killing of GMBs, lipid peroxidation etc.), sensitive to canonical ferroptosis inhibitors such as ferrostatin and liproxstatin?
- Fig. 5D/H: Similarly, would ferroptosis inhibitors blunt these effects? Moreover, experiments with genetic deletion of MPO would be important to strengthen the proposed cellular process.
- What is the mechanism of MPO transfer? Does it involve cell-cell contact?
- What is the impact of neutrophils on tumor vascularization of implanted tumors?
- Fig 6: GPX4 overexpression or KO of ACSL4 prolongs survival of tumor bearing mice. How would reduced tumor necrosis allow increased overall survival? Are there any differences in tumor size? Are the number of other cell types in the brain such as microglia, astrocytes and endothelial cells altered?
- Fig. S6: The authors should provide immunoblot data on ACSL4 – why does exogenously expressed GPX4 run at almost 30 kDa?
- P9/Fig 8: It remains unclear what the sequences of events are – would hypoxia first induce non-ferroptotic cell death, followed by neutrophil recruitment, ferroptosis, DMAP release etc. which in turn triggers a vicious cycle? Again, what would be the contribution of other (immune) cells of the brain in such a scenario? It is likely that ferroptosis may also contribute to the very first event as hypoxia/reoxygenation is known to induce ferroptosis and GPX4 was frequently found to be downregulated/lost under these conditions.

Reviewer #3 (Remarks to the Author): with expertise in neutrophils

The manuscript “Neutrophil-induced ferroptosis promotes tumor necrosis in glioblastoma” explores the mechanism underlying necrosis formation in GBM. Yee and colleagues utilize a constitutively activated TAZ to model the mesenchymal subtype of GBM. They show that in this model, the tumors develop extensive necrotic regions which are heavily infiltrated by neutrophils. They provide various evidence to demonstrate that neutrophils play a detrimental role in the process of tumor necrosis and that they kill tumor cells via ferroptosis. Importantly, the authors associated this process, of ferroptosis mediated tumor necrosis with increased tumor aggressiveness. Finally, the authors provide evidence which suggest a correlation between neutrophil infiltration, tumor necrosis and poor survival in patients with GBM. Overall, the study explores an important topic and is well executed. Furthermore, it provides a novel angle for tumor necrosis which was thus far mostly attributed to vascular insufficiency. That said, there are several issues that should be addressed before this manuscript can be accepted for publication.

Major

1. In figure 2 the authors show that Ly6G+ neutrophils in necrotic TAZ(4SA) and state that they are rarely found in control tumors. Does it mean that necrotic areas in control GBM are devoid of neutrophils? If so, the authors should better explain the scope of relevance of their findings.
2. TAZ overactivation in this model generates extensive necrosis. When looking at control tumors – is TAZ activity increased specifically in the PNZ?
3. The activation of TAZ generates necrotic tumors and shortens overall survival. The authors should explain why it is. Do tumors grow larger? Do they proliferate more? Do they spread faster?
4. In figure 4H the authors compare CD45+ and CD45- cells. The appropriated experiment should compare CD45-, CD45 Ly6G- and CD45 Ly6G+ cells. This comparison would show not only the difference between tumor cells and immune cells but would also show that within the immune cells it is specifically neutrophils.
5. The authors use 3 human cell lines to justify their findings. While this indeed strongly supports their claims it limits the experiments to immune deficient mice. My specific concern is that since this study explores functions of immune cells in the context of GBM using an immune deficient setting. There is evidence to suggest that neutrophil function in nude mice differs from that of neutrophils in immune competent mice. The authors should therefore demonstrate that their key observations are also relevant in an immune competent settings.

Minor

1. Line 167, bottom of page 5 – top of page 6 “ also examine their temporal connection.” Not clear what the authors wanted to say.
2. Line 204 – Ly6 should be changed to Ly6G.
3. Line 268 – the authors mention 3 types of neutrophils but figure S4A shows only two.
4. Line 288 – “Necrotic regions had stronger.....”. The authors claims are not clearly evident from the figure.
5. Line 419 – there is only one panel in figure S7.
6. The data from the coculture experiments demonstrating neutrophil cytotoxicity are presented as % of survival. While this is perfectly fine - showing the opposite – i.e. the % of cells dying – would be more impressive visually.

REVIEWER COMMENTS

Reviewer #1 (Remarks to the Author): with expertise in glioblastoma

Major aspects

1. The selection of cell lines used should be explained. As the molecular background (p53 status etc.) may be critical for the phenomenon observed, these details need to be provided and evaluated.

We have explained the selection of cell lines used and provided the details of p53 status in the cell lines on pages 4 and 5.

2. The different genetic constructs for TAZ are well chosen.

We thank the reviewer for appreciation of our experimental design.

3. The change between labeling the infiltrating cells neutrophils and broader CD11b+CD45+ myeloid cells should be made clearer. Also in the analysis of the temporal correlation of tumor recruitment of inflammatory cells.

We have rewritten these parts on pages 5 and 6 to make them clearer.

4. As it has been reported that tumors are attractive to myeloid/progenitor/stem cells, it should be closer carved out, which cells are present at which stage in which quantities.

We have performed new experiments to characterize cells of the myeloid lineage during tumor development. First, we used two myeloid cell markers, CD11b and CD45, to examine myeloid cells in LN229^{TAZ(4SA)} tumors at different stages of tumor progression. Flow cytometry indicated that CD45⁺ cells (i.e. infiltrating mouse immune cells) in tumors at day 20 after tumor implantation can be separated into three major populations based on CD11b and CD45 signal intensities, which we named CD11b^{high}CD45^{high}, CD11b^{med}CD45^{med}, and CD11b^{low}CD45^{low} cells (Figure S2A). At this stage, the tumor-infiltrating immune cells consist of nearly equal proportions of the three cell populations. As tumors grow, the CD11b^{high}CD45^{high} cells gradually become the dominant population (Figures S2A and S2B). Previous studies reported that microglia in inflamed brains can be distinguished from peripherally-infiltrating macrophages based on lower microglial CD45 expression¹. However, CD45 expression in neutrophils relative to microglia and macrophages in the brain was unclear. To examine which cell population contains neutrophils, we used the murine neutrophil marker Ly6G. The CD11b^{high}CD45^{high} population largely consisted of Ly6G⁺ cells, whereas the other two populations essentially lack Ly6G⁺ cells (Figure S2A). Such specific enrichment of Ly6G⁺ cells does not change during tumor development (Figure S2C). A description of these results was added on page 6.

5. The time courses for the depletion experiments in Figure 3 are not fully transparent. Early/late is well understood, but the respective times are not matching.

We have provided data of individual tumors in Figure S3E. The tumor collection date for each one was shown in this figure to increase transparency.

6. The conclusion that a reduced necrotic area in Lys6-treated mice is due to reduced neutrophil numbers has an alternative, that is a direct impact of Lys6 on the necrosis development (i) by local effects on the resident immune cells or (ii) alternative tumor cell-directed mechanism.

We agree with the reviewer that one cannot exclude a possible effect on the resident immune cells, such as microglia, although it was reported that the anti-Ly6G antibody that we used can specifically recognize neutrophils but not monocytes/macrophages². We have added this potential caveat in discussion, page 18.

To examine if the Ly6G antibody has a direct impact on tumor cells, we applied the antibody directly in the culture of tumor cells. The new result showed that the anti-Ly6G antibody by itself does not affect tumor cell proliferation *in vitro* (Figure S3A). Therefore, it is unlikely that the antibody can directly affect tumor cells. A description of these results was added on page 7.

7. Data on the cytotoxicity of neutrophils from different species are mainly correlative, but not sufficiently proven.

The bioluminescence approach has been used to study neutrophil-induced tumor cell death³⁻⁶. As an alternative approach, we used the colony formation assay, a commonly used method to evaluate the consequence of cell death. To alleviate the reviewer's concern, we have employed a third approach, the Sytox fluorescent dye cell viability assay, which can more definitively detect cell death based on monitoring cell membrane integrity. This method has been widely used to study cell death, including ferroptosis. Results from this new method have been added in Figures S3H, S4D and S5E. A description of these results was added on pages 8 and 10.

8. The reference for the survival rates as reported in Fig 4a,b are not fully clear. It seems that this presentation has been chosen to avoid full transparency on the actual numeric values. Are the data from one experiment and if so, absolute values and respective error bars would be a preferred mode of presentation.

We have replaced the normalized results with the actual numeric values from the assays in Figures 3C and 4A. We have added the actual numeric values for Figures 4B and S4C in Figure S4A.

9. Similar, as the experiments (in vitro and ex vivo) in 4 a and 4b are different, appropriate ways to depict the different data are valuable.

As stated in #8 above, we have added the actual numeric values Figures 4A and 4B to reflect the different data.

10. Data on MPO transfer and inhibition in Figures 5/6 are convincing.

We thank the reviewer for appreciation of the convincing data.

11. The clinical data are meant to transfer the experimental findings to the patient site. Whereas the correlation between neutrophil signatures and tumor subtypes are interesting (see below), the attempt to correlated pretumor diagnosis circulating neutrophil numbers with tumor/necrosis ratio is prone to false conclusions due to the mobility of the compartment, impact of infections, medication (steroid plus/minus) etc. The used concept of tumor subtypes, which is commonly used relies on the stability and ignores the multiple subtypes in one summary subtype as published in a very elegant study form the MGH.

We agree with the reviewer that pre-operative circulating neutrophil numbers (as used in our study) may be affected by infections, medications (e.g. +/- steroids) etc. Given that electronic medical records are not yet centralized in our country, we were unable to trace and pinpoint the exact dates on which peri-operative steroids were started for each patient to stratify by steroid use in our cohort. This is indeed a potential source of confounding. We have added a note of this caveat on page 17. We also agree with the reviewer that one tumor may be heterogeneous in subtypes. This is the inherent caveat when using the TCGA dataset, which only examined one sample for each tumor and assigned tumor subtype based on this. The Ivy Glioblastoma Atlas Project examined multiple samples in each tumor based on sub-tumoral localization, which is related to necrosis. We have used this dataset, which may alleviate such a concern to some extent. We have added notes about these limitations on page 15.

12. The proposed clinical applicability of the findings would need some more consideration of the appropriate timing of an intervention in the real world setting.

Necrosis usually appears along with tumor progression. Considering this timing factor, we have revised the proposal as “when advanced-stage tumors manifest necrosis, targeting ferroptotic cell death might benefit GBM patients by curtailing tumor necrosis-triggered sequelae” on page 20.

Minor aspects

1. TAZ needs to be explained in the abstract.

We have provided the full name of TAZ in the abstract.

Reviewer #2 (Remarks to the Author): with expertise in ferroptosis

Tumor necrosis is commonly observed in poorly vascularized solid tumors, although the mechanisms how this is brought about remain poorly understood. In this work, the authors studied whether tumor infiltrating neutrophils contribute to the necrotic core by triggering ferroptosis in glioblastoma cells, both in cultured cells and intracranially implanted tumors. The authors provide evidence that the transfer of myeloperoxidase (MPO) from neutrophils to tumor cells may be one of the underlying mechanisms how ferroptotic cell death determines the extent of tumor necrosis, survival of tumor bearing mice and GBM patient prognosis. While the study is of interest, there are a number of uncertainties and open questions that call for a careful revision.

Comments:

- Fig. 1: The authors should include immunoblot data on exogenously expressed TAZ

We have included immunoblot data on exogenously expressed TAZ(4SA) and TAZ(4SA)-S51A in Figure S1A.

- Fig. 3/4: Are some of the effects (e.g. neutrophil mediated killing of GBMs, lipid peroxidation etc.), sensitive to canonical ferroptosis inhibitors such as ferrostatin and liproxstatin?

Yes, neutrophil mediated killing of GBM cells can be inhibited by canonical ferroptosis inhibitors, including ferrostatin and liproxstatin-1 (Figures 4B, S4A, and S4C). In addition, it can also be inhibited by the iron chelator, deferoxamine (DFO) (Figures 4B and S4A). We confirmed these observations by performing a second cell viability assay (Figure S4D). We also examined lipid peroxidation and found that neutrophil-induced lipid peroxidation in GBM cells can also be inhibited by these inhibitors (Figures S4G and S4H). A description of these new results is on page 10.

- Fig. 5D/H: Similarly, would ferroptosis inhibitors blunt these effects? Moreover, experiments with genetic deletion of MPO would be important to strengthen the proposed cellular process.

We have performed additional experiments and found that the reduced viability of PKH26^{high} tumor cells can be rescued by ferrostatin-1, liproxstatin-1, or DFO (Figure S5E). A description of these new results is on pages 11 and 12.

We have performed experiments with genetic depletion of MPO. To examine if MPO is necessary for neutrophil-induced tumor cell death, it was knocked down from differentiated HL-60 and 32Dcl3 cells by two different shRNAs (Figures S5H and S5I). These MPO-depleted cells were significantly less capable of inducing tumor cell death than controls (Figure 5I), concordant with results from pharmacological inhibition as in Figure 5H. A description of these new results is on page 13.

- What is the mechanism of MPO transfer? Does it involve cell-cell contact?

To understand whether such neutrophil-derived MPO transfer into tumor cells required cell-cell contact, we employed two approaches. First, tumors were cultured in conditioned media (CM) derived from neutrophil monoculture. Second, tumor cells and neutrophils were cocultured with a cell-impermeable membrane in between, with tumor cells on the bottom of the wells and neutrophils above the membrane. In both conditions, barely any MPO⁺ puncta were observed in tumor cells when compared to those cocultured with neutrophils directly (Figures S5F and S5G). These results together suggested that direct cell-cell contact or close proximity between tumor cells and neutrophils is necessary for intercellular MPO transfer. A description of these new results is on page 12.

- What is the impact of neutrophils on tumor vascularization of implanted tumors?

We have performed additional experiments and found that intratumoral endothelial (CD31⁺) cells appeared to be less abundant when neutrophils were depleted (Figure S3F). This observation is consistent with the notion that TANs promote tumor angiogenesis of implanted tumors ^{7, 8}. A description of these new results is on page 8.

- Fig 6: GPX4 overexpression or KO of ACSL4 prolongs survival of tumor bearing mice. How would reduced tumor necrosis allow increased overall survival? Are there any differences in tumor size? Are the number of other cell types in the brain such as microglia, astrocytes and endothelial cells altered?

When examining the effect of GPX4 overexpression or ACSL4 knockdown on tumor necrosis, we used tumors with similar sizes. To evaluate tumor growth, we have provided new results recording tumor growth. Expression of rGPX4 slightly slowed tumor growth, although the difference was statistically insignificant (Figure S6D). Tumor growth was dampened when ACSL4 expression was more effectively silenced (Figure S6I, sh#41). However, when ACSL4 was partially depleted, there was little inhibitory effect on tumor growth (Figures 6I, sh#42). These results suggested that reduced tumor growth may not wholly explain prolonged survival. A description of these results has been added on page 14.

We performed new experiments to examine the impact on the GBM tumor microenvironment when ferroptosis is inhibited by the above two genetic manipulations. It appeared that numbers of CD31⁺ endothelial cells and GFAP⁺ astrocytes were not affected (Figures S6J and S6K). Interestingly, when examined by TMEM119, a murine microglia marker, we saw fewer intratumoral microglia in tumors expressing rGPX4 or ACSL4 shRNAs. In contrast, brain parenchymal microglia showed no difference (Figures S6J and S6K). A description of these results has been added on page 15.

While it is still unclear how necrosis can decrease overall survival in both pre-clinical and clinical GBMs, our results found that five cytokines were more abundant in LN229^{TAZ(4SA)} (with necrosis) than LN229^{vector} (without necrosis) tumors (Figure 6F). Most of these cytokines were associated with necrosis and poorer survival in human GBMs. Some cytokines, such as IL8 and IL6, have been implicated in promoting GBM progression. Moreover, IL6 has been implicated to cause cachexia and systemic deterioration ⁹. With what we have observed in our model, we suspect that secretion of these pro-tumorigenic factors and/or recruitment and activation of other

immune cells (e.g. microglia) by ferroptosed tumor cells may lead to certain non-autonomous tumor effects that precipitate neuroinflammation (or more precisely “necroinflammation”) and cerebral cytokine storm, eventually leading to irreversible cerebral edema, cachexia, multi-organ dysfunction, and death. We have added this discussion on pages 15 and 20.

- Fig. S6: The authors should provide immunoblot data on ACSL4 – why does exogenously expressed GPX4 run at almost 30 kDa?

We have performed the immunoblotting and included the new immunoblot data on ACSL4 in Figure S6F.

Exogenously expressed GPX4 (rGPX4, arrow) contains a mitochondria-targeting sequence¹⁰ and appears to be larger than endogenous GPX4 (asterisk), which likely does not have this sequence. We have added this explanation on page 13.

- P9/Fig 8: It remains unclear what the sequences of events are – would hypoxia first induce non-ferroptotic cell death, followed by neutrophil recruitment, ferroptosis, DMAP release etc. which in turn triggers a vicious cycle? Again, what would be the contribution of other (immune) cells of the brain in such a scenario? It is likely that ferroptosis may also contribute to the very first event as hypoxia/reoxygenation is known to induce ferroptosis and GPX4 was frequently found to be downregulated/lost under these conditions.

We agree with the reviewer that these are interesting questions following up our discovery in this manuscript. It was traditionally thought that hypoxia is the cause of tumor necrosis. Our studies revealed neutrophil-triggered ferroptosis is a new component involved in necrosis formation. We tried to trace back to the very early stage of necrosis development. It seems that neutrophil accumulation can be seen in certain areas where tissues appear to be slightly damaged (Figure 2G, day 16). This result in combination with the traditional view of the contribution of hypoxia agrees with the reviewer’s thought that hypoxia first induces cell death or damage, followed by neutrophil recruitment. Whether hypoxia-induced cell death is ferroptosis is worth further study. Our model suggests that once neutrophil-triggered ferroptosis occurs, it would induce a positive feedback loop of necrosis formation. Our studies cannot exclude the contribution of other immune cells. Although this is an interesting direction, we respectfully suggest that this may be out of the scope of the current manuscript. We have added comments regarding this direction on page 18.

Reviewer #3 (Remarks to the Author): with expertise in neutrophils

The manuscript “Neutrophil-induced ferroptosis promotes tumor necrosis in glioblastoma” explores the mechanism underlying necrosis formation in GBM. Yee and colleagues utilize a constitutively activated TAZ to model the mesenchymal subtype of GBM. They show that in this model, the tumors develop extensive necrotic regions which are heavily infiltrated by neutrophils. They provide various evidence to demonstrate that neutrophils play a detrimental role in the process of tumor necrosis and that they kill tumor cells via ferroptosis. Importantly, the authors associated this process, of ferroptosis mediated tumor necrosis with increased tumor aggressiveness. Finally, the authors provide evidence which suggest a correlation between neutrophil infiltration, tumor necrosis and poor survival in patients with GBM. Overall, the study explores an important topic and is well executed. Furthermore, it provides a novel angle for tumor necrosis which was thus far mostly attributed to vascular insufficiency. That said, there are several issues that should be addressed before this manuscript can be accepted for publication.

Major

1. In figure 2 the authors show that Ly6G⁺ neutrophils in necrotic TAZ(4SA) and state that they are rarely found in control tumors. Does it mean that necrotic areas in control GBM are devoid of neutrophils? If so, the authors should better explain the scope of relevance of their findings.

The control LN229^{vector} tumors do not develop detectable necrosis (Figures 1F and S1C). Some LN229^{TAZ(4SA-S51A)} tumors can develop small necrosis. However, LN229^{TAZ(4SA-S51A)} tumors are much more similar to LN229^{vector} than LN229^{TAZ(4SA)} tumors. We have added these clarifications on page 5. Therefore, there are no necrotic areas devoid of neutrophils.

2. TAZ overactivation in this model generates extensive necrosis. When looking at control tumors – is TAZ activity increased specifically in the PNZ?

As clarified in the above comment #1, there is no necrosis in control tumors. Therefore, there is no PNZ either. Because of this, we are unable to evaluate if TAZ activity is increased specifically in the PNZ.

3. The activation of TAZ generates necrotic tumors and shortens overall survival. The authors should explain why it is. Do tumors grow larger? Do they proliferate more? Do they spread faster?

We have provided the growth curve of LN229^{vector} and LN229^{TAZ(4SA)} tumors (Figure S1B). It shows that LN229^{TAZ(4SA)} tumors grow faster than LN229^{vector} tumors. However, at endpoints, LN229^{vector} tumors do grow to comparable sizes, often even larger, than LN229^{TAZ(4SA)} tumors, as measured by BLI (Figure S1B). The difference in their growth rates may be the primary reason that LN229^{TAZ(4SA)} tumor-bearing mice exhibited significantly shorter overall survival. Overall, our studies did not seem to support that tumor size is the major cause of necrosis. LN229^{TAZ(4SA)} tumors develop necrosis as early as day 16 after implantation (Figure 2G). At this time, tumor sizes are still relatively small. In contrast, LN229^{vector} tumors do not develop necrosis even at sizes that are much larger than LN229^{TAZ(4SA)} tumors (Figure 2G vs. Figure

S1C). This indicates that tumor sizes are not critical for developing necrosis, and this is thus consistent with clinical observations that small GBMs can also develop necrosis¹¹. Although the exact instigator of necrosis in LN229^{TAZ(4SA)} tumors remains to be determined, the foci which appear to be necrotic niches are infiltrated by neutrophils (Figure 2G). These neutrophils may contribute to necrosis formation during an early stage of necrosis development (i.e. neo-necrosis). Since neutrophil depletion cannot completely eliminate necrosis formation (Figure 3A), our results cannot exclude contributions by other factors, such as ischemia, to necrosis induction. Nevertheless, the results did support that neutrophils play a remarkable role in promoting necrosis development to its fullest extent. On the other hand, activation of TAZ is able to promote ferroptotic cell death under canonical ferroptosis stimuli¹². Therefore, TAZ activation may sensitize GBM cells to ferroptosis. We have added the related discussions on pages 17 and 18.

4. In figure 4H the authors compare CD45+ and CD45- cells. The appropriated experiment should compare CD45-, CD45 Ly6G- and CD45 Ly6G+ cells. This comparison would show not only the difference between tumor cells and immune cells but would also show that within the immune cells it is specifically neutrophils.

The rationale of the experiment in Figure 4H is: We found that necrotic regions had stronger Liperfluo signals than cellular tumor regions (Figure 4G, red versus white asterisks). Interestingly, peri-necrotic zones exhibited the strongest Liperfluo signals (Figure 4G, arrows). Because tumor cells and neutrophils are the two major populations in these peri-necrotic zones (Figure 2A), we wondered whether it was neutrophils or tumor cells that contributed to the higher Liperfluo signals. To address this issue, we designed the experiment in Figure 4H. Specifically, because neutrophils are leukocytes, we used CD45, a murine leukocyte marker, to distinguish immune cells from tumor cells (Figure S4H). The result of this experiment allowed us to find out that tumor cells (CD45⁻) showed much stronger Liperfluo signals than immune cells (CD45⁺) (Figures 4H and S4J). These results indicated that tumor cells in peri-necrotic zones contained higher levels of intracellular lipid hydroperoxides than tumor cells in other locations or non-tumor cells. Although we agree with the reviewer that comparing the level of lipid hydroperoxides in neutrophils with other immune cells would be interesting, because this was not our purpose, we respectively suggest that adding the marker of Ly6G⁺ to specifically study neutrophils may be out of the scope of this study.

5. The authors use 3 human cell lines to justify their findings. While this indeed strongly supports their claims it limits the experiments to immune deficient mice. My specific concern is that since this study explores functions of immune cells in the context of GBM using an immune deficient setting. There is evidence to suggest that neutrophil function in nude mice differs from that of neutrophils in immune competent mice. The authors should therefore demonstrate that their key observations are also relevant in an immune competent settings.

We agree with the reviewer that an immunocompetent mouse model is more ideal to further examine our observations. While many genetically engineered mouse glioma models and syngeneic mouse glioma lines have been reported, as far as we know, these reported models have not shown such extensive tumor necrosis (>30% of tumor size, Figures 1E and 1F). Necrosis shown in the published models usually appears to be small. Therefore, these models may not be feasible to study the development of extensive necrosis found in human GBM. To alleviate the reviewer's concern, we tested the GL261 syngeneic mouse glioma cell line (a commonly used, publicly available syngeneic line).

[Redacted]

Although it is still premature to speculate whether there is a difference between mouse and human glioma cells in sensitivity to ferroptosis, this observation suggested that necrosis formation may depend on specific genetic background and/or tumor microenvironment. Given this knowledge, we respectfully suggest that using other unknown models may be out of the scope of the current manuscript. We have noted such caveats on the mouse models in the discussion, on page 18. Although our mouse models are limited to an immunodeficient setting, we have cross-examined the key observations, such as the link of neutrophils, ferroptosis and necrosis in human GBM (Figures 7 and 8). These suggested that our findings are also relevant in immunocompetent settings.

Minor

1. Line 167, bottom of page 5 – top of page 6 “ also examine their temporal connection.” Not clear what the authors wanted to say.

We have clarified by rewriting the part on page 6.

2. Line 204 – Ly6 should be changed to Ly6G.

We have made the change on page 8.

3. Line 268 – the authors mention 3 types of neutrophils but figure S4A shows only two.

We have corrected the text into “two” on page 10.

4. Line 288 – “Necrotic regions had stronger.....”. The authors claims are not clearly evident from the figure.

We have reworded the sentence on page 10.

5. Line 419 – there is only one panel in figure S7.

We have corrected the text into S7A on page 15.

6. The data from the coculture experiments demonstrating neutrophil cytotoxicity are presented as % of survival. While this is perfectly fine - showing the opposite – i.e. the % of cells dying – would be more impressive visually.

We appreciate the reviewer's suggestion. Because of the concerns of Reviewer 1, comment #8, we have provided the actual numeric values from the assays instead of plotting in this way.

Reference:

1. Sedgwick, J.D. *et al.* Isolation and direct characterization of resident microglial cells from the normal and inflamed central nervous system. *Proc Natl Acad Sci U S A* **88**, 7438-7442 (1991).
2. Daley, J.M., Thomay, A.A., Connolly, M.D., Reichner, J.S. & Albina, J.E. Use of Ly6G-specific monoclonal antibody to deplete neutrophils in mice. *J Leukoc Biol* **83**, 64-70 (2008).
3. Karimi, M.A. *et al.* Measuring cytotoxicity by bioluminescence imaging outperforms the standard chromium-51 release assay. *PLoS One* **9**, e89357 (2014).
4. Sagiv, J.Y. *et al.* Phenotypic diversity and plasticity in circulating neutrophil subpopulations in cancer. *Cell Rep* **10**, 562-573 (2015).
5. Granot, Z. *et al.* Tumor entrained neutrophils inhibit seeding in the premetastatic lung. *Cancer Cell* **20**, 300-314 (2011).
6. Fu, X. *et al.* A simple and sensitive method for measuring tumor-specific T cell cytotoxicity. *PLoS One* **5**, e11867 (2010).
7. Deryugina, E.I. *et al.* Tissue-infiltrating neutrophils constitute the major in vivo source of angiogenesis-inducing MMP-9 in the tumor microenvironment. *Neoplasia* **16**, 771-788 (2014).
8. Zhou, S.L. *et al.* CXCL5 contributes to tumor metastasis and recurrence of intrahepatic cholangiocarcinoma by recruiting infiltrative intratumoral neutrophils. *Carcinogenesis* **35**, 597-605 (2014).
9. Narsale, A.A. & Carson, J.A. Role of interleukin-6 in cachexia: therapeutic implications. *Curr Opin Support Palliat Care* **8**, 321-327 (2014).
10. Savaskan, N.E., Ufer, C., Kuhn, H. & Borchert, A. Molecular biology of glutathione peroxidase 4: from genomic structure to developmental expression and neural function. *Biol Chem* **388**, 1007-1017 (2007).
11. Raza, S.M. *et al.* Necrosis and glioblastoma: a friend or a foe? A review and a hypothesis. *Neurosurgery* **51**, 2-12; discussion 12-13 (2002).
12. Yang, W.H. *et al.* The Hippo Pathway Effector TAZ Regulates Ferroptosis in Renal Cell Carcinoma. *Cell Rep* **28**, 2501-2508 e2504 (2019).

REVIEWER COMMENTS

Reviewer #1 (Remarks to the Author):

My comments are well addressed and the additional Experiments well executed.

Reviewer #2 (Remarks to the Author):

Congrats to the authors as they have done a fantastic job and addressed all of my comments (except one) in a very careful and diligent way.

Unfortunately, one major issue remains with the human GPX4 sequence used for overexpression (see my previous comment on the incorrect size of GPX4!). It seems they used the wrong variant (nuclear GPX4 or mitochondrial GPX4) for the studies as only the cytosolic variant is important for mouse survival (see Jiang et al JBC 2009) and ferroptosis protection, while nuclear and mitochondrial forms only play important roles during sperm development (see Conrad et al MCB 2005; Schneider et al FASEB J 2009)!

Mitochondrial GPX4 contains a classical mitochondrial targeting signal (MTS) at the N-terminus, but this variant is only expressed in spermatocytes (unlike other rather shaky reports) and translocates to the mitochondrial matrix. Upon translocation in the matrix in sperm cells, the MTS is cleaved, meaning there is now more a difference in size detectable between "cytosolic/somatic" GPX4 and mitochondrial GPX4 in testis.

Nuclear GPX4 is encoded by a separate exon and renders the protein approx. 30-34 kD in size. This form has an N-terminal nuclear localization signal for nuclear translocation but most importantly this - like the mitochondrial one - does not rescue GPX4 KO cells from ferroptosis.

In either case, the authors attached the Flag-HA tag to the N-terminus and thus there might be wrong location of human GPX4 in the cells? On the other hand, it could well be that this tag masks recognition of either the nuclear or mitochondrial form used (which does not get clear from the vendors' page), so it would still be at the right location in the best case scenario. We know that the N-terminus allows larger tags (see Mannes et al FASEB 2011), without impinging on the functionality of "cytosolic" GPX4.

As such, I urge the authors to double check what variant they actually expressed in their studies. The "cytosolic" one starts with "MCASRDDWRCARS", the mitochondrial one with "MSLGRLCRLK KPALL...."and nuclear one with "MGRAGAGSPG RRRQRCQSRG RRRP....". There are a lot of misannotations and confusions in public databases and vendors usually refer to this. So please clarify.

Reviewer #3 (Remarks to the Author):

Going over the revised version of the manuscript Yee and colleagues I it clearly evident that it is much improved. The authors made a sincere, and mostly successful, attempt to address the reviewers' concerns. However, a few issues still remain.

Major

1. In my comment #3 I raised a concern regarding the reason TAZ activation reduces overall survival and whether this is linked to tumor necrosis. The authors demonstrate that the tumors grow faster but find that tumor size is not the reason for tumor necrosis. So. As the authors

conclude – tumor growth per se is the reason mice with activated TAZ tumors succumb earlier. This leaves us without an explanation for why tumors with TAZ activation grow faster and takes us back to my original question – Do these tumors proliferate more, apoptose less or spread faster?

2. In my comment #5 I expressed a concern regarding the use of immune deficient mice. Although the authors tried addressing this concern using a mouse cell line (which should be absolutely sufficient) this concern persists. Specifically, because this cell line did not induce necrotic tumors even when expressing the active form of TAZ. This raises additional concerns. At this point we have three tumors that follow the mechanism suggested by the authors and one that doesn't. The fact that the one that doesn't is the only one tested in immune competent mice is disturbing. What if these observations are only valid in immune deficient mice where neutrophils behave differently? If the phenomenon described by the authors is indeed strain-dependent how wide is the scope of the findings? The analogy to the findings in human GBM is indeed supportive of the authors' claims but is not sufficient to prove that the same mechanism is relevant in immune competent organisms.

Minor

1. Regarding my comment #4 – although it seems like a reasonably simple experiment the authors prefer giving an elaborate explanation for why they use CD45 and not Ly6G as a marker. Since this is the case, they should include a clear statement indicating that this is the contribution of CD45+ cells and not necessarily neutrophils.

REVIEWER COMMENTS

Reviewer #1 (Remarks to the Author):

My comments are well addressed and the additional Experiments well executed.

Reviewer #2 (Remarks to the Author):

Congrats to the authors as they have done a fantastic job and addressed all of my comments (except one) in a very careful and diligent way.

Unfortunately, one major issue remains with the human GPX4 sequence used for overexpression (see my previous comment on the incorrect size of GPX4!). It seems they used the wrong variant (nuclear GPX4 or mitochondrial GPX4) for the studies as only the cytosolic variant is important for mouse survival (see Jiang et al JBC 2009) and ferroptosis protection, while nuclear and mitochondrial forms only play important roles during sperm development (see Conrad et al MCB 2005; Schneider et al FASEB J 2009)!

Mitochondrial GPX4 contains a classical mitochondrial targeting signal (MTS) at the N-terminus, but this variant is only expressed in spermatocytes (unlike other rather shaky reports) and translocates to the mitochondrial matrix. Upon translocation in the matrix in sperm cells, the MTS is cleaved, meaning there is now more a difference in size detectable between “cytosolic/somatic” GPX4 and mitochondrial GPX4 in testis.

Nuclear GPX4 is encoded by a separate exon and renders the protein approx. 30-34 kD in size. This form has an N-terminal nuclear localization signal for nuclear translocation but most importantly this - like the mitochondrial one - does not rescue GPX4 KO cells from ferroptosis.

In either case, the authors attached the Flag-HA tag to the N-terminus and thus there might be wrong location of human GPX4 in the cells? On the other hand, it could well be that this tag masks recognition of either the nuclear or mitochondrial form used (which does not get clear from the vendors' page), so it would still be at the right location in the best case scenario. We know that the N-terminus allows larger tags (see Mannes et al FASEB 2011), without impinging on the functionality of “cytosolic” GPX4.

As such, I urge the authors to double check what variant they actually expressed in their studies. The “cytosolic” one starts with “MCASRDDWRCARS ...”, the mitochondrial one with “MSLGRLCRLK KPALL...” and nuclear one with “MGRAGAGSPG RRRQRCQSRG RRRP...”. There are a lot of misannotations and confusions in public databases and vendors usually refer to this. So please clarify.

We thank the reviewer for pointing out the issue regarding using the recombinant human GPX4 (rGPX4). We have verified the sequence and confirmed that the rGPX4 is the mitochondrial form starting with “MSLGRLCRLKALL...”. To address the reviewer’s concern, we have examined the localization of rGPX4 in LN229^{TAZ(4SA)} cells via immunocytochemistry. Ectopically expressed rGPX4 appeared to be diffuse throughout in the cytosol and not limited to the mitochondria (Figure S6B). This observation is consistent with the reviewer’s prediction that rGPX4 is actually cytosolically localized, presumably due to the attached Flag-HA tag or that mitochondrial protein transporters may be overwhelmed by rGPX4. Therefore, this may explain why we still saw the rescuing effects by expressing rGPX4 in LN229^{TAZ(4SA)}. A description of this result was added on page 13.

Reviewer #3 (Remarks to the Author):

Going over the revised version of the manuscript Yee and colleagues I it clearly evident that it is much improved. The authors made a sincere, and mostly successful, attempt to address the reviewers’ concerns. However, a few issues still remain.

Major

1. In my comment #3 I raised a concern regarding the reason TAZ activation reduces overall survival and whether this is linked to tumor necrosis. The authors demonstrate that the tumors grow faster but find that tumor size is not the reason for tumor necrosis. So. As the authors conclude – tumor growth per se is the reason mice with activated TAZ tumors succumb earlier. This leaves us without an explanation for why tumors with TAZ activation grow faster and takes us back to my original question – Do these tumors proliferate more, apoptose less or spread faster?

We apologize for not fully addressing the previous question of this reviewer. LN229^{TAZ(4SA)} tumors develop remarkable necrosis. However, these tumors still grow faster than LN229^{vector} tumors. This result indicates that the tumor bulk not only needs to compensate for necrosis-caused cell loss, but also to gain extra cells. Therefore, it is very likely that enhanced proliferation is the main reason for faster growth of the tumors.

2. In my comment #5 I expressed a concern regarding the use of immune deficient mice. Although the authors tried addressing this concern using a mouse cell line (which should be absolutely sufficient) this concern persists. Specifically, because this cell line did not induce necrotic tumors even when expressing the active form of TAZ. This raises additional concerns. At this point we have three tumors that follow the mechanism suggested by the authors and one that doesn’t. The fact that the one that doesn’t is the only one tested in immune competent mice is disturbing. What if these observations are only valid in immune deficient mice where neutrophils behave differently? If the phenomenon described by the authors is indeed strain-

dependent how wide is the scope of the findings? The analogy to the findings in human GBM is indeed supportive of the authors' claims but is not sufficient to prove that the same mechanism is relevant in immune competent organisms.

We appreciate and fully understand the reviewer's concern regarding the use of immune deficient mice. To the best of our knowledge, none of the reported syngeneic immunocompetent mouse GBM models and xenograft immunodeficient mouse GBM models fully recapitulate the extensive tumor necrosis seen in human GBM. This situation has been a hurdle to studying GBM-associated necrosis. Therefore, our knowledge of the mechanism driving such necrosis is scarce. The mouse model reported in the current study has provided an unprecedented opportunity to study GBM-associated necrosis. Although the findings from this immunodeficient model cannot be directly tested in an immunocompetent model due to the current technique limitation, our human GBM cross-examination provides translational value and suggests that our results could be extrapolatable to immunocompetent human patients. The GL261 result that we provided in Revision 1 was intriguing. Although it may suggest a potentially different mechanism in an immunocompetent setting, the result may also be due to differences in genetic background and cell metabolism between the three human GBM cell lines that we used and this particular mouse GBM cell line. Therefore, we respectively suggest cautious interpretation of this GL261 result and sincerely ask for the reviewer's understanding of our current technique limitation. Nevertheless, we completely agree that the mechanism needs to be further examined in immunocompetent models when they are available.

Minor

1. Regarding my comment #4 – although it seems like a reasonably simple experiment the authors prefer giving an elaborate explanation for why they use CD45 and not Ly6G as a marker. Since this is the case, they should include a clear statement indicating that this is the contribution of CD45⁺ cells and not necessarily neutrophils.

We have reworded the sentence indicating that they are CD45⁺ immune cells on page 10.

REVIEWERS' COMMENTS

Reviewer #2 (Remarks to the Author):

While the authors have shown that rGPX4 is evenly distributed throughout the cell, a functional readout for instance by expressing rGPX4 in inducible GPX4 knockout cells would have been more appropriate as cytosolic/short form of GPX4 also localizes to the mitochondrial intermembrane space (see "Short form glutathione peroxidase 4 is the essential isoform required for survival and somatic mitochondrial functions.

Liang H, Yoo SE, Na R, Walter CA, Richardson A, Ran Q. J Biol Chem. 2009 Nov 6;284(45):30836-44").

Nonetheless, it seems that the HA tag masks the MTS which would otherwise lead to localization of the protein in the mitochondrial matrix. At least, the paper mentioned in the foregoing and the one by Mannes et al (see "Cysteine mutant of mammalian GPx4 rescues cell death induced by disruption of the wild-type selenoenzyme. Mannes AM, Seiler A, Bosello V, Maiorino M, Conrad M. FASEB J. 2011 Jul;25(7):2135-44") should be commented as over expression of rGPX4 might be a limitation of this study.

REVIEWERS' COMMENTS

Reviewer #2 (Remarks to the Author):

While the authors have shown that rGPX4 is evenly distributed throughout the cell, a functional readout for instance by expressing rGPX4 in inducible GPX4 knockout cells would have been more appropriate as cytosolic/short form of GPX4 also localizes to the mitochondrial intermembrane space (see “Short form glutathione peroxidase 4 is the essential isoform required for survival and somatic mitochondrial functions. Liang H, Yoo SE, Na R, Walter CA, Richardson A, Ran Q. J Biol Chem. 2009 Nov 6;284(45):30836-44”).

Nonetheless, it seems that the HA tag masks the MTS which would otherwise lead to localization of the protein in the mitochondrial matrix. At least, the paper mentioned in the foregoing and the one by Mannes et al (see “Cysteine mutant of mammalian GPx4 rescues cell death induced by disruption of the wild-type selenoenzyme. Mannes AM, Seiler A, Bosello V, Maiorino M, Conrad M. FASEB J. 2011 Jul;25(7):2135-44”) should be commented as over expression of rGPX4 might be a limitation of this study.

We agree with the reviewer that over expression of rGPX4 might be a limitation of this study and expressing rGPX4 in inducible GPX4 knockout cells would have been more appropriate. We have cited the two papers with comments on rGPX4 on page 20.